# Timestep Rescheduling in Diffusion Inversion

**Shangquan Sun** [1] [*]  **Ting Gong** [2] [*]  **Zhirui Liu** [1]  **Jiamin Wu** [3]  **Runkai Zhao** [4]  **Mianxin Liu** [5]
**Wenqi Ren** [1]  **Xiaochun Cao** [1]

## Abstract

Diffusion inversion, which maps images back to the Gaussian latent space of a diffusion model, is a critical task for image reconstruction and editing. While DDIM enables fast deterministic inversion, it inherently introduces deviations that accumulate into noticeable inversion errors. Existing methods often address this by solving a fixed-point problem but largely overlook how the selection of the diffusion timestep in the noise scheduler influences inversion fidelity. In this work, we reveal that the deviation scale in diffusion inversion is strongly dependent on the timestep size, and exhibits a parabolic trend, with larger errors concentrated at both small and large timesteps. Based on this finding, we propose a simple yet effective nonuniform timestep scheduler that integrates a global rescaling with a local dynamic programming based rescheduling, enabling a strategic allocation of computational effort that minimizes the overall inversion error and preserves higher inversion accuracy. Our method serves as an off-the-shelf enhancement for existing inversion techniques and requires no extra parameters or computational overhead. Through extensive experiments, we verify that integrating our scheduler consistently boosts the performance of existing inversion methods, achieving superior results in image reconstruction and editing.

## 1. Introduction

The emergence of large-scale text-to-image diffusion models has transformed the landscape of controllable image gen-

[*]Equal contribution First author: Shangquan Sun <shangquansun@gmail.com>. [1]School of CyberScience and Technology, Sun Yat-sen University, Shenzhen Campus, Shenzhen 518107, China [2]Tsinghua University [3]CUHK [4]University of Sydney [5]Shanghai AI Laboratory. Correspondence to: Wenqi Ren <rwq.renwenqi@gmail.com>.

*Proceedings of the 43rd International Conference on Machine Learning*, Seoul, South Korea. PMLR 306, 2026. Copyright 2026 by the author(s).

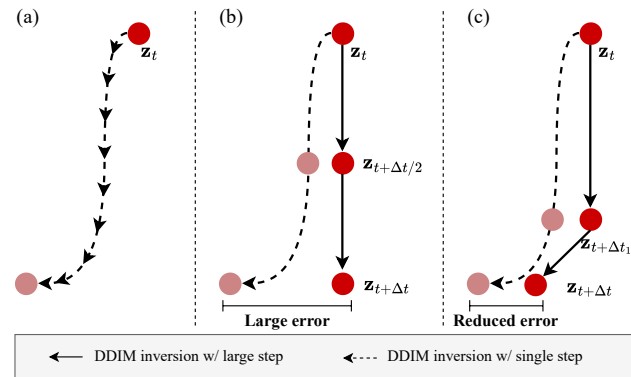

*Figure 1.* Illustration of the impact of timestep scheduling on diffusion inversion. (a) Inversion with many small fine-grained steps achieves high accuracy but incurs high computational cost. (b) Uniformly spaced timesteps accelerate inversion but accumulate significant reconstruction errors. (c) Our adaptive rescheduling strategy assigns larger steps to low error regions and finer steps to high error regions, balancing efficiency and accuracy.

eration (Ho et al., 2020; Ramesh et al., 2022; Rombach et al., 2022; Saharia et al., 2022), enabling impressive progress in both synthesis and semantic manipulation (Alaluf et al., 2024; Avrahami et al., 2023; 2022; Brooks et al., 2023; Ge et al., 2023; Hertz et al., 2023; Kawar et al., 2023; Meng et al., 2022; Parmar et al., 2023; Patashnik et al., 2023; Tumanyan et al., 2023; Bai & Melas-Kyriazi, 2024; Luo et al., 2023; Nichol & Dhariwal, 2021). A key enabler for applying these generative models to real image editing is diffusion inversion, which is the process of mapping a given real image $\mathbf{z}_0$ back to a latent noise vector $\mathbf{z}_T$ such that the diffusion model can faithfully reconstruct the original image through its denoising trajectory. This inversion process is fundamentally challenging because diffusion models are trained for a backward generation direction, i.e., progressively denoising latent variables toward clean images, while the reverse mapping from image to noise is not explicitly learned.

Due to this asymmetry, recovering an accurate latent representation for an arbitrary real image remains an ill-posed problem, especially since real images may not lie perfectly within the model's training distribution (Han et al., 2024; Mokady et al., 2023; Huberman-Spiegelglas et al., 2024;

Miyake et al., 2025; Garibi et al., 2024). To address this, Denoising Diffusion Implicit Model (DDIM) (Song et al., 2021) introduced a deterministic and efficient formulation that enables approximate inversion by analytically reversing the denoising step. However, this approximation inevitably introduces reconstruction errors, leading to distortions in the reconstructed image, particularly in few-step diffusion settings with coarse temporal discretization. Recent works have sought to mitigate these errors by refining the inversion equation via fixed-point iterations (Meiri et al., 2023; Hang et al., 2024; Garibi et al., 2024; Samuel et al., 2025), showing improvements but still facing limitations in accuracy and computational efficiency.

Despite these advances, one important yet underexplored factor in diffusion inversion is the *timestep scheduling*. Existing studies primarily focus on reducing local inversion errors within each step but often overlook how the distribution and spacing of diffusion timesteps influence the overall reconstruction fidelity. While the impact of timestep scheduling has been investigated in diffusion sampling (Xue et al., 2024; Zheng et al., 2024; 2025; Xia et al., 2024) and training (Wang et al., 2025), its role in the inversion process remains largely unknown. Most existing inversion pipelines still adopt uniform timestep sampling from $0$ to $T$, potentially suboptimal for accurate latent recovery.

To this end, we systematically investigate the impact of timestep scheduling in diffusion inversion and propose an adaptive scheme that effectively reduces global inversion errors across different diffusion depths. Our analysis reveals that, even with large time intervals, the inversion error can be reformulated as a scaled fixed-point problem, where the scaling coefficient depends on the magnitude of the timestep and the discrepancy between consecutive model predictions. While previous studies (Meiri et al., 2023; Hang et al., 2024; Garibi et al., 2024) have examined fixed-point formulations for inversion, the role of this scaling factor, that is intrinsically tied to the chosen timestep interval, has not been explored. Since the selection of a current timestep influences the subsequent trajectory, finding an optimal scheduling strategy remains nontrivial. We begin by analyzing the error trajectory with respect to timestep intervals and observe a clear trend: larger step sizes introduce higher reconstruction errors, aligning with intuitive expectations. Furthermore, for a fixed timestep, the error exhibits a parabolic relationship with the target position, i.e., decreasing rapidly at smaller diffusion steps but growing sharply at larger ones. Motivated by this observation, we first introduce a coarse rescheduling mechanism that adjusts the allocation of timesteps, stretching them globally to mitigate accumulated error. To further refine this process, we formulate a dynamic programming–based optimization that adaptively redistributes timesteps in a fine-grained manner, effectively minimizing cumulative inversion error. A diagrammatic comparison between our method and the previous uniform timestep schedule is presented in Fig. 1. Through extensive experiments on both image reconstruction and image editing tasks, we demonstrate that our timestep rescheduling strategy significantly improves the inversion quality of existing diffusion-based methods, especially in few-step diffusion settings. Notably, our approach introduces no additional computational overhead and can be seamlessly integrated into current diffusion inversion pipelines.

Our main contributions are as follows:

- We derive a theoretical formulation of the inversion error induced by timestep jumps and reveal its connection to a scaled fixed-point problem.

- We propose a coarse-to-fine timestep rescheduling framework that adaptively allocates larger steps in low-error regions to minimize overall inversion error.

- We validate our approach through extensive experiments, demonstrating consistent improvements across various existing inversion baselines in both image reconstruction and image editing without increasing computational cost.

**Conflict of Interest Disclosure.** The authors declare no financial conflicts of interest related to this work.

## 2. Related Works

**Image Synthesis and Editing** is central to computer vision. Early works focused on generative adversarial networks (GANs) (Goodfellow et al., 2014; Karras et al., 2019; Brock et al., 2019). However, GANs often exhibit training instability and limited generalization to complex controls, such as natural language instructions or multi-modal conditioning. Recent diffusion-based methods enable controllable, high-fidelity synthesis guided by text, sketches, or other conditions (Sohl-Dickstein et al., 2015; Ho et al., 2020; Dhariwal & Nichol, 2021). Text-to-image diffusion models (Rombach et al., 2022; Saharia et al., 2022; Ramesh et al., 2022; Balaji et al., 2022; Esser et al., 2024; Couairon et al., 2023; Kulikov et al., 2025; Yoon et al., 2025; Dong et al., 2023) generate images from noise conditioned on text, narrowing the gap between intent and visual creation. Yet, consistent, precise identity-preserving edits under text guidance remain challenging.

**Diffusion Models** are now the dominant paradigm for generative modeling due to stable training and high sample quality. They are also used in 3D reconstruction (Jin et al., 2024; Wu et al., 2024a), image restoration (Xia et al., 2023; Yue et al., 2025), and beyond. They iteratively denoise a Gaussian sample into an image, modeling the data distribution in a tractable manner. Denoising Diffusion Probabilistic Models

(DDPMs) (Ho et al., 2020) and Latent Diffusion Models (LDMs) (Rombach et al., 2022) are representative frameworks. DDIM (Song et al., 2021) introduced deterministic sampling for faster generation and feasible inversion. Later works improved efficiency via acceleration (Karras et al., 2022), dynamic thresholding (Saharia et al., 2022), or few-step distillation (Salimans & Ho, 2022). These advances enable as few as 4–8 denoising steps with high fidelity, supporting interactive editing and content creation.

**Diffusion Inversion** aims to recover the latent noise associated with a real image, enabling bidirectional mapping between image and latent spaces. A widely used deterministic method is DDIM inversion (Song et al., 2021), which approximates the forward diffusion with linearized dynamics. This approximation can cause reconstruction degradation, especially in few-step diffusion where errors accumulate across timesteps. To mitigate this, numerous improved inversion methods have been proposed (Mokady et al., 2023; Wallace et al., 2023; Hong et al., 2025; Bao et al., 2025; Li et al., 2025; Zuo et al., 2026). Null-text inversion (Mokady et al., 2023) optimizes the empty-prompt embedding to better align forward and reverse processes. Miyake et al. (Miyake et al., 2025) extend this by optimizing text embeddings, improving fidelity but reducing editing flexibility. EDICT (Wallace et al., 2023) uses invertible affine coupling layers for an exact mapping, while BDIA (Zhang et al., 2024) offers a more efficient approximation. Other work relies on refinement or optimization: TurboEdit (Wu et al., 2024b) uses an encoder-guided iterative scheme for few-step models, and DirectInv (Ju et al., 2024) decouples source/target branches to better preserve content. Fixed-point and implicit-function approaches—AIDI (Pan et al., 2023), ReNoise (Garibi et al., 2024), GNRI (Samuel et al., 2025), EasyInv (Zhang et al., 2025), and ExactDPM (Hong et al., 2024)—boost fidelity by explicitly solving the DDIM equations. Scheduler design has also been explored to reduce accumulated error (Lin et al., 2024a;b). In particular, Schedule Your Edit (Lin et al., 2024a) redesigns the diffusion noise schedule for image editing, whereas our method keeps the underlying scheduler family fixed and reallocates the discrete timesteps under a fixed inference budget. Exact and reversible inversion methods such as EDICT, BDIA, and ExactDPM replace the inversion formulation or solver to improve reconstruction fidelity; TRDI instead acts as a scheduler-level retrofit for existing deterministic inversion pipelines and is complementary to these directions. Meanwhile, stochastic inversion methods (Huberman-Spiegelglas et al., 2024; Brack et al., 2024; Deutch et al., 2024) revisit DDPM-style formulations for near-exact reconstructions, but typically require many steps and large latent storage, limiting interactive use. Despite these advances, the role of timestep selection in inversion is still underexplored. We study this factor systematically and propose an adaptive scheduling mechanism that improves reconstruction and editing.

## 3. Method

### 3.1. Preliminaries

**Diffusion Models** are generative models that learn to synthesize data by reversing a progressive noising process that transforms clean samples into Gaussian noise through a forward Markov chain and reconstructs them via a learned reverse process. Formally, given an image latent $\mathbf{z}_0$, the forward diffusion process adds Gaussian noise step-by-step to produce intermediate latents $\{\mathbf{z}_t\}_{t=1}^{T}$:

$$q(\mathbf{z}_t|\mathbf{z}_{t-1}) = \mathcal{N}(\mathbf{z}_t; \sqrt{a_t}\mathbf{z}_{t-1}, (1-a_t)\mathbf{I}), \qquad (1)$$

where $a_t \in (0,1)$ is a noise schedule controlling the amount of noise added at step $t$. After $T$ steps, the data distribution approaches a standard Gaussian, i.e., $\mathbf{z}_T \sim \mathcal{N}(\mathbf{0}, \mathbf{I})$.

The reverse process aims to recover $\mathbf{z}_{t-1}$ from $\mathbf{z}_t$ by gradually removing noise. It is parameterized by a neural network $\epsilon_\theta(\mathbf{z}_t, t, p)$ conditioned on a prompt $p$ (e.g., text embedding):

$$p_\theta(\mathbf{z}_{t-1}|\mathbf{z}_t, p) = \mathcal{N}(\mathbf{z}_{t-1}; \mu_\theta(\mathbf{z}_t, t, p), \sigma_t^2\mathbf{I}), \qquad (2)$$

where $\mu_\theta(\mathbf{z}_t, t, p)$ is computed to predict the noise-free component and $\sigma_t^2$ is the noise variance. Sampling from this process produces realistic images from Gaussian noise.

**Deterministic Schedulers.** Different from the original DDPM that is stochastic, DDIM (Song et al., 2021) interprets the reverse diffusion as solving an ordinary differential equation (ODE), allowing for non-stochastic sampling:

$$\mathbf{z}_{t-1} = \sqrt{\frac{\alpha_{t-1}}{\alpha_t}}\mathbf{z}_t - \sqrt{\alpha_{t-1}}\,\Delta\psi(\alpha_t, 1)\,\epsilon_\theta(\mathbf{z}_t, t, p), \quad (3)$$

where we have $\alpha_t = \prod_{i=1}^{t} a_i$, $\psi(\alpha) = \sqrt{1/\alpha - 1}$, and $\Delta\psi(\alpha_t, \Delta t) = \psi(\alpha_t) - \psi(\alpha_{t-\Delta t})$. This deterministic mapping enables faster sampling while maintaining high fidelity. Alternative schedulers such as Euler and Heun methods (Karras et al., 2022) reformulate the denoising process by modeling the latent evolution through a predicted velocity function.

**Diffusion Inversion** aims to compute the latent noise $\mathbf{z}_T$ corresponding to a given image latent $\mathbf{z}_0$, such that the denoising trajectory starting from $\mathbf{z}_T$ reconstructs $\mathbf{z}_0$. In the context of deterministic schedulers such as DDIM, inversion can be viewed as finding a latent $\mathbf{z}_t$ that satisfies the implicit relation:

$$\mathbf{z}_t = \sqrt{\frac{\alpha_t}{\alpha_{t-1}}}\mathbf{z}_{t-1} + \sqrt{\alpha_t}\,\Delta\psi(\alpha_t, 1)\,\epsilon_\theta(\mathbf{z}_t, t, p). \qquad (4)$$

Because the component $\epsilon_\theta(\mathbf{z}_t, t, p)$ of this equation is implicit in $\mathbf{z}_t$, a common practice is to approximate $\mathbf{z}_t$ with $\mathbf{z}_{t-1}$ in the network input, i.e.,

$$\mathbf{z}_t \approx \sqrt{\frac{\alpha_t}{\alpha_{t-1}}}\mathbf{z}_{t-1} + \sqrt{\alpha_t}\,\Delta\psi(\alpha_t, 1)\,\epsilon_\theta(\mathbf{z}_{t-1}, t-1, p), \quad (5)$$

which defines the standard DDIM inversion (Song et al., 2021). This approximation allows efficient inversion with negligible computational overhead, but it introduces an error at each step that accumulates across timesteps, leading to imperfect reconstruction and degraded editing fidelity (Mokady et al., 2023; Pan et al., 2023; Wallace et al., 2023).

### 3.2. Diffusion Inversion Error w.r.t Timesteps

To reduce accumulated approximation errors, recent studies (Meiri et al., 2023; Pan et al., 2023; Garibi et al., 2024; Samuel et al., 2025) employ iterative refinement based on fixed-point iteration (Rhoades, 1976), repeatedly updating the latent until convergence. However, most of these methods focus on minimizing local approximation errors, with little attention given to the impact of timestep selection. Instead of sampling (Zheng et al., 2024; 2025; Xue et al., 2024) and training (Wang et al., 2025) of diffusion model, diffusion inversion requires balancing text-based modification with pixel-level reconstruction accuracy. Here, we present an analysis of how timestep selection influences the diffusion inversion error.

Similar to Eq. 4, the inversion process with a large timestep $\Delta t$ can be generalized as follows:

$$\mathbf{z}_t^{(\Delta t)} = \sqrt{\frac{\alpha_t}{\alpha_{t-\Delta t}}}\mathbf{z}_{t-\Delta t} + \sqrt{\alpha_t}\,\Delta\psi(\alpha_t, \Delta t)\,\epsilon_\theta(\mathbf{z}_t, t, p). \tag{6}$$

The special case of $\Delta t = 1$ reduces to Eq. 4. Assuming that single-step ($\Delta t = 1$) inversion achieves the most accurate latent reconstruction compared to large-step inversion, the additional inversion error introduced by varying timestep sizes can be defined as the discrepancy between the results of large-step and single-step inversions,

$$\delta(\mathbf{z}_t, t, \Delta t) = \|\mathbf{z}_t^{(\Delta t)} - \mathbf{z}_t^{(1)}\|$$
$$= \Bigg\| \frac{\sqrt{\alpha_t}}{\sqrt{\alpha_{t-\Delta t}}} \left[\mathbf{z}_{t-\Delta t} + \sqrt{\alpha_{t-\Delta t}}\,\Delta\psi(\alpha_t, \Delta t)\,\epsilon_\theta(\mathbf{z}_t, t, p)\right] \tag{7}$$
$$- \frac{\sqrt{\alpha_t}}{\sqrt{\alpha_{t-1}}} \left[\mathbf{z}_{t-1} + \sqrt{\alpha_{t-1}}\,\Delta\psi(\alpha_t, 1)\,\epsilon_\theta(\mathbf{z}_t, t, p)\right] \Bigg\|.$$

Given a discrete time interval $\Delta t > 1$, the corresponding large-step diffusion forward process can be formulated as a Gaussian transition $q(\mathbf{z}_{t-1}|\mathbf{z}_{t-\Delta t})$ that moves the latent from $\mathbf{z}_{t-\Delta t}$ to $\mathbf{z}_{t-1}$, and it can be parameterized as

$$q(\mathbf{z}_{t-1}|\mathbf{z}_{t-\Delta t}) : \mathbf{z}_{t-1} \leftarrow \mathbf{z}_{t-1}^{(\Delta t-1)}$$
$$= \frac{\sqrt{\alpha_{t-1}}}{\sqrt{\alpha_{t-\Delta t}}}\mathbf{z}_{t-\Delta t} + \sqrt{\alpha_{t-1}}\,\Delta\psi(\alpha_{t-1}, \Delta t - 1)\,\epsilon_\theta(\mathbf{z}_{t-1}, t-1, p).$$

By substituting this expression in Eq 7, the inversion error with respect to timesteps can be expressed as a scaled fixed-point problem:

$$\delta(\mathbf{z}_t, t, \Delta t)$$
$$= \| \underbrace{c_{\boldsymbol{\alpha}}(t, \Delta t)}_{\text{scaling coefficient}} \underbrace{(\epsilon_\theta(\mathbf{z}_t, t, p) - \epsilon_\theta(\mathbf{z}_{t-1}, t-1, p))}_{\Delta\epsilon_\theta:\text{ single-step fixed-point problem}} \|, \quad (8)$$

where the scaling coefficient

$$c_{\boldsymbol{\alpha}}(t, \Delta t) = \sqrt{\alpha_t}\Delta\psi(\alpha_{t-1}, \Delta t - 1), \tag{9}$$

depends solely on the noise and timestep schedules, and $\boldsymbol{\alpha} = \{\alpha_t\}_{t=1}^T$ denotes the discretized noise schedule parameters. The remaining term $\Delta\epsilon_\theta$ represents a local fixed-point problem that has been extensively investigated in previous studies (Song et al., 2021; Garibi et al., 2024; Zhang et al., 2025; Samuel et al., 2025). The complete derivations and a discussion of commonly used noise schedulers are provided in Sec. A and B, respectively. Assuming the diffusion model is well trained such that its output follows a standard Gaussian distribution, the scale of the error coefficient can be treated as the error magnitude itself because $\Delta\epsilon_\theta$ also follows a standard Gaussian distribution. Based on the analysis of Eq. 9, we next present our strategy for adjusting $\Delta t$ and the corresponding timesteps.

This view is also consistent with the ODE interpretation of deterministic diffusion inversion. The DDIM trajectory can be regarded as a discretization of a probability-flow ODE, where nonuniform timesteps change the local discretization defect accumulated along the inversion path. Thus, rather than proposing a new ODE solver, TRDI reallocates a fixed step budget according to the scheduler-dependent surrogate $c_{\boldsymbol{\alpha}}$, playing the role of an adaptive discretization signal for existing inversion pipelines.

### 3.3. Timestep Rescheduling in Diffusion Inversion

Suppose we have a total of $K$ steps, with the selected timesteps denoted as $\{t_k\}_{k=1}^K$. The commonly used selection strategies for timesteps are detailed in Sec. B, where all adopt uniformly distributed timesteps. Our objective is to adaptively reschedule arbitrary timestep inputs such that the accumulated error is minimized.

We first visualize $c_{\boldsymbol{\alpha}}(t, \Delta t)$ with respect to $t$ for different step sizes $\Delta t$ in Fig. 2. It can be observed that larger steps induce greater inversion errors. Additionally, a parabolic relationship emerges between error and timestep index: when the step index is small, the error coefficient is large and rapidly decreases, but it rises again as the step index approaches the end. Based on this observation, the intuition is to avoid large steps when the inversion error is high and to use fewer but larger steps when the error scale is small. Additionally, since errors accumulated during the early steps

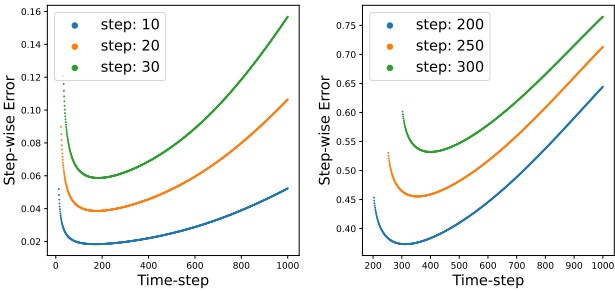

*(a) The case of finer steps*    *(b) The case of larger steps*

*Figure 2.* Illustration of the timestep-related inversion error coefficient $c_{\boldsymbol{\alpha}}(t, \Delta t)$, where different curves correspond to different $\Delta t$ values. Two key observations can be made: (1) larger step sizes lead to greater errors, and (2) the error is initially high at small step indices, rapidly decreases, and then gradually increases again as the timestep index approaches $T$.

*Table 1.* Comparison of image reconstruction performance on the MSCOCO dataset (Lin et al., 2014) using the diffusion model SD v1.5 (Rombach et al., 2022). The improvement gains in percentage are highlighted in red.

| Method | PSNR↑ | SSIM$_{\times 10^2}$ ↑ | LPIPS$_{\times 10^3}$ ↓ |
|--------|-------|------------|-------------|
| DDIM | 20.07 | 65.11 | 193.97 |
| w/ Ours | 20.21$_{0.70\%}$ | 65.73$_{0.95\%}$ | 187.85$_{3.16\%}$ |
| ReNoise | 22.35 | 69.46 | 166.27 |
| w/ Ours | 22.67$_{1.43\%}$ | 70.42$_{1.38\%}$ | 157.30$_{5.39\%}$ |
| NPI | 20.82 | 66.22 | 182.01 |
| w/ Ours | 21.08$_{1.25\%}$ | 67.05$_{1.25\%}$ | 175.41$_{3.63\%}$ |
| GNRI | 22.14 | 69.72 | 147.02 |
| w/ Ours | 22.32$_{0.81\%}$ | 70.39$_{0.96\%}$ | 141.33$_{3.87\%}$ |

can significantly affect the model output $\epsilon_\theta(\mathbf{z}_t, t, p)$ in later stages with larger timesteps, it is crucial to maintain high sensitivity during the early stage.

**Global Rescaling of Timestep Schedule.** Therefore, we propose a global adjustment of the original timesteps that slightly stretches all timesteps dynamically in preparation for subsequent fine-grained rescheduling. Given a uniformly spaced timestep input

$$t_k = t_1 + (t_K - t_1)\frac{k-1}{K-1}, \ \forall k = \{1, ...K\}, \quad (10)$$

we present a reformulation to rescale each timestep:

$$t_k = t_1 + (t_K - t_1)\left(\frac{k-1}{K-1}\right)^\gamma, \ \forall k = \{1, ...K\}, \quad (11)$$

where $\gamma$ is a scaling hyperparameter that controls the stretching of timesteps. When $\gamma = 1$, no rescaling is applied; when $\gamma > 1$, timesteps expand toward earlier steps; and when $\gamma < 1$, they become denser toward the end. The optimal value of $\gamma$ is confirmed through ablation studies.

---

**Algorithm 1** TRDI: Timestep Rescheduling in Diffusion Inversion

**Require:** number of inversion steps $K$, noise schedule $\{\alpha_t\}_{t=1}^T$ where $T$ is the number of training diffusion steps, original timesteps $\{t_k\}_{k=1}^K$, window width $d$, scaling hyperparameter $\gamma$

**Ensure:** Rescheduled inversion timesteps $\{\hat{t}_k\}_{k=1}^K$

1: **Timestep Rescaling:**
2: Update each $t_k$ by Eq. 11.
3: **Dynamic Programming:**
4: Initialize accumulative error map $\mathbf{E} \in \mathbb{R}^{K \times T}$.
5: Initialize intermediate timestep recorder $\mathbf{R} \in \mathbb{R}^{K \times T}$.
6: **for** each step $k \in \{1, \ldots, K\}$ **do**
7:     **for** each timestep $t \in \{t_k - d, \ldots, t_k + d\}$ **do**
8:         **if** $k = 1$ **then**
9:             Update $\mathbf{E}[1, t]$ using Eq. 12.
10:         **else**
11:             Update $\mathbf{E}[k, t]$ using Eq. 13.
12:             Record the $\arg\min$ of Eq. 13 in $\mathbf{R}[k, t]$.
13:         **end if**
14:     **end for**
15: **end for**
16: **Retrieve timesteps:**
17: $\hat{t}_K \leftarrow \min\{\mathbf{E}[K, t]\}_{t=t_K-d}^{t_K+d}.$
18: **for** each step $k \in \{K-1, \ldots, 1\}$ **do**
19:     $\hat{t}_k \leftarrow \mathbf{R}[k, \hat{t}_{k+1}].$
20: **end for**
21: **return** $\{\hat{t}_k\}_{k=1}^K.$

---

**Local Rescheduling via Dynamic Programming.** The previous rescaling of timesteps provides only a coarse adjustment and does not have direct access to the exact error term in Eq. 9. To this end, we introduce a windowed dynamic programming approach that locally refines the timestep schedule. Given a sliding window of length $2d + 1$ centered on the current step, we define a cost map $\mathbf{E}$ that records the minimum accumulated error across the sliding window. Specifically, $\mathbf{E}[k, t]$ represents the minimum cost of reaching step index $t$ at $k$-th step. The initialization of $\mathbf{E}$ is expressed as

$$\mathbf{E}[1, t] = c_{\boldsymbol{\alpha}}(t, t), \ \forall t \in \{t_1 - d, ..., t_1 + d\}, \quad (12)$$

where $\mathbf{E}[1, t]$ stores the inversion error from the input image $\mathbf{z}_0$ to the latent representation at the current step, covering for all possible step indices within the window centered at $t_1$. The recursive update of the dynamic programming formulation for subsequent steps is defined as

$$\mathbf{E}[k, t] = \min\{\mathbf{E}[k-1, h] + c_{\boldsymbol{\alpha}}(t, t-h)\}_{h=t_{k-1}-d}^{t_{k-1}+d}, \\ \forall t \in \{t_k - d, ..., t_k + d\}, k \in \{2, ..., K\}, \quad (13)$$

where the recursive update searches for the locally optimal transition between adjacent timesteps within the defined

*Table 2.* Comparison of diffusion inversion methods on the image editing benchmark PIE-Bench (Ju et al., 2024). The diffusion models used are SDXL (Podell et al., 2023) and SDXL Turbo (Sauer et al., 2024). The improvement gains in percentage are indicated in red.

| Model | Method | Structure Distance$_{\times 10^3}$ ↓ | PSNR ↑ | Background Preservation LPIPS$_{\times 10^3}$ ↓ | MSE$_{\times 10^4}$ ↓ | SSIM$_{\times 10^2}$ ↑ | CLIP Similariy Whole ↑ | Edited ↑ |
|---|---|---|---|---|---|---|---|---|
| SDXL | DDIM | 19.43 | 26.26 | 89.24 | 39.94 | 86.27 | 24.15 | 20.98 |
|  | w/ Ours | 15.63$_{24.31\%}$ | 26.53$_{1.02\%}$ | 84.20$_{5.99\%}$ | 37.89$_{5.41\%}$ | 86.60$_{0.38\%}$ | 24.19$_{0.17\%}$ | 21.09$_{0.52\%}$ |
|  | ReNoise | 27.81 | 25.70 | 99.84 | 49.35 | 84.02 | 24.10 | 21.03 |
|  | w/ Ours | 27.64$_{0.62\%}$ | 25.81$_{0.43\%}$ | 98.81$_{1.04\%}$ | 47.09$_{4.80\%}$ | 84.78$_{0.90\%}$ | 24.07$_{0.12\%}$ | 21.09$_{0.28\%}$ |
|  | NPI | 19.43 | 26.26 | 89.04 | 39.91 | 86.28 | 24.11 | 20.98 |
|  | w/ Ours | 16.36$_{18.77\%}$ | 26.54$_{0.94\%}$ | 83.13$_{7.11\%}$ | 37.74$_{5.75\%}$ | 86.66$_{0.44\%}$ | 24.21$_{0.41\%}$ | 21.10$_{0.57\%}$ |
|  | GNRI | 41.66 | 21.84 | 139.45 | 96.51 | 79.75 | 25.54 | 21.79 |
|  | w/ Ours | 39.82$_{4.62\%}$ | 22.21$_{1.67\%}$ | 138.19$_{0.91\%}$ | 88.14$_{9.50\%}$ | 80.19$_{0.55\%}$ | 25.45$_{0.35\%}$ | 21.82$_{0.14\%}$ |
| SDXL Turbo | DDIM | 85.55 | 18.36 | 185.10 | 198.04 | 66.07 | 25.64 | 22.40 |
|  | w/ Ours | 70.64$_{17.43\%}$ | 19.03$_{3.65\%}$ | 166.52$_{11.16\%}$ | 170.54$_{16.13\%}$ | 68.37$_{3.36\%}$ | 25.79$_{0.58\%}$ | 22.78$_{1.67\%}$ |
|  | ReNoise | 69.48 | 19.11 | 197.75 | 540.68 | 70.01 | 25.31 | 22.83 |
|  | w/ Ours | 63.55$_{9.33\%}$ | 20.13$_{5.07\%}$ | 180.68$_{9.45\%}$ | 516.25$_{4.73\%}$ | 72.03$_{2.80\%}$ | 25.29$_{0.08\%}$ | 22.91$_{0.35\%}$ |
|  | NPI | 35.01 | 21.81 | 124.96 | 93.26 | 74.88 | 25.11 | 22.14 |
|  | w/ Ours | 31.98$_{9.47\%}$ | 22.33$_{2.33\%}$ | 119.86$_{4.25\%}$ | 83.69$_{11.44\%}$ | 75.86$_{1.29\%}$ | 24.97$_{0.56\%}$ | 22.00$_{0.64\%}$ |
|  | GNRI | 32.06 | 22.18 | 124.92 | 88.48 | 75.52 | 24.42 | 21.46 |
|  | w/ Ours | 23.63$_{35.67\%}$ | 23.39$_{5.17\%}$ | 110.13$_{13.43\%}$ | 67.76$_{30.58\%}$ | 77.53$_{2.59\%}$ | 24.43$_{0.04\%}$ | 21.54$_{0.37\%}$ |

*Table 3.* The ablation study evaluates the global rescaling factor $\gamma$ and the window width $d$ used in dynamic programming. The diffusion model employed is SDXL (Podell et al., 2023), with a total of 50 timesteps ($\Delta t = 20$). The input timesteps follow the "leading" scheduling strategy. The baseline inversion method used for comparison is the DDIM inversion (Song et al., 2021). The best and second-best results in both the upper and lower halves of the table are highlighted in **bold** and underlined, respectively.

| Settings $\gamma$ | $d$ | Structure Distance$_{\times 10^3}$ ↓ | PSNR ↑ | Background Preservation LPIPS$_{\times 10^3}$ ↓ | MSE$_{\times 10^4}$ ↓ | SSIM$_{\times 10^2}$ ↑ | CLIP Similariy Whole ↑ | Edited ↑ |
|---|---|---|---|---|---|---|---|---|
| 1.10 | 0 | 17.33 | 26.26 | 98.16 | 40.51 | 85.74 | 24.03 | **21.03** |
| 1.05 | 0 | **17.07** | **26.45** | 88.49 | **38.75** | 86.36 | **24.18** | **21.03** |
| 1.00 | 0 | 19.43 | 26.26 | 89.24 | 39.94 | 86.27 | 24.15 | 20.98 |
| 0.90 | 0 | 23.11 | 26.14 | **85.31** | 41.60 | **86.37** | 23.90 | 20.63 |
| 1.05 | 0 | 17.07 | 26.45 | 88.49 | 38.75 | 86.36 | 24.18 | 21.03 |
| 1.05 | 2 | 17.05 | 26.26 | 93.66 | 40.55 | 85.93 | 24.18 | **21.11** |
| 1.05 | 5 | 16.39 | 26.38 | 89.25 | 39.28 | 86.25 | 24.18 | 21.09 |
| 1.05 | 8 | **15.63** | 26.53 | 84.20 | 37.89 | 86.60 | **24.19** | 21.09 |
| 1.05 | 10 | 15.68 | **26.84** | **77.06** | **35.51** | **87.17** | 24.13 | 20.94 |

window. Consequently, each entry $\mathbf{E}[k, t]$ records the local minimum accumulated error cost at step $k$ and timestep index $t$. The complete procedure of the proposed timestep rescheduling method is summarized in Algorithm 1.

### 3.4. Discussion of Timestep Rescheduling

One may observe that since all $a_t$ values from $0$ to $T$ are known, it is theoretically possible to apply dynamic programming directly to obtain the global minimum of overall error. However, the derivation relies on approximating the error coefficient as the error itself, based on the premise that a well-trained diffusion model maintains consistent performance across all timesteps, such that the differences between consecutive model outputs follow a standard Gaus-

sian distribution, which may not always hold (Wang et al., 2025). Performing global dynamic programming to minimize the error coefficient could lead to a biased timestep distribution, as the exact value of the remaining term $\Delta \epsilon_\theta$ in Eq. 8 is inherently unknown. To address these issues, we thus adopt a coarse-to-fine strategy that adaptively reschedules the timesteps, achieving minimization of the error coefficient while preserving a relatively uniform distribution of the original timesteps. This analysis is therefore local and surrogate-based, rather than an exact characterization of the final global inversion error. The surrogate may be less reliable when model mismatch, semantic editing conflicts, or higher-order solver effects dominate the reconstruction error. In such cases, TRDI is best viewed as a lightweight scheduler-level retrofit that can complement, but not replace,

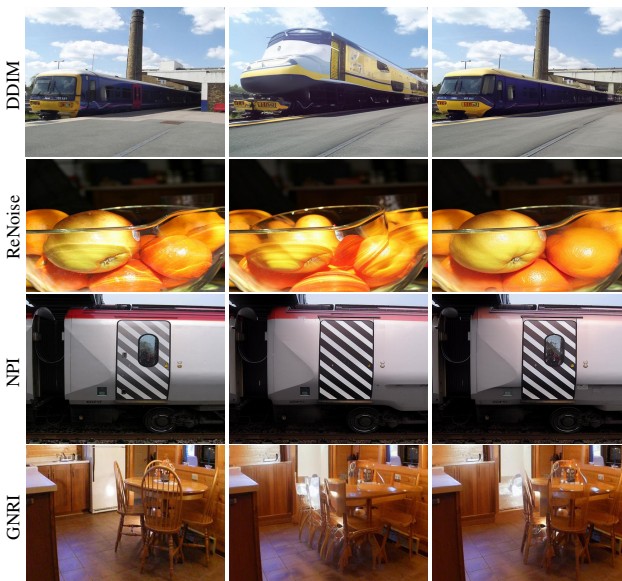

*(a)* Input      *(b)* Baseline      *(c)* w/ Ours

*Figure 3.* Qualitative comparison of image reconstruction results before and after applying our timestep rescheduling method across different diffusion inversion baselines on SD v1.5 (Rombach et al., 2022) using a total of 50 steps.

stronger inversion formulations or solvers.

## 4. Experiments

### 4.1. Experimental Settings

In this section, we conduct extensive experiments to validate the effectiveness of our proposed method. We evaluate its performance in terms of both reconstruction quality and editability, with a particular emphasis on the improvements achieved by existing inversion methods when integrated with our inversion timestep rescheduling strategy.

**Baselines.** We extensively evaluate the impact of our method on the performance of existing diffusion inversion methods, including DDIM inversion (Song et al., 2021), ReNoise (Garibi et al., 2024), Negative Prompt Inversion (NPI) (Miyake et al., 2025), and GNRI (Samuel et al., 2025).

**Datasets.** For the reconstruction task, we use the validation set of the MSCOCO dataset (Lin et al., 2014) and employ reference captions as prompts (Li et al., 2023; Chen et al., 2015). Following prior works (Zhang et al., 2025; Miyake et al., 2025), we randomly sample 1,000 image-caption pairs from MSCOCO and use the first of the five candidate captions as the prompt. For the image editing task, we evaluate our method on PIE-Bench (Ju et al., 2024), a comprehensive benchmark for prompt-based image editing. It contains 700 images across 10 distinct editing types, with each sample annotated with source and target prompts, an editing instruction, edit subjects, and a precise editing mask for accurate metric computation.

"a woman in a ~~jacket~~ blouse standing in the rain"

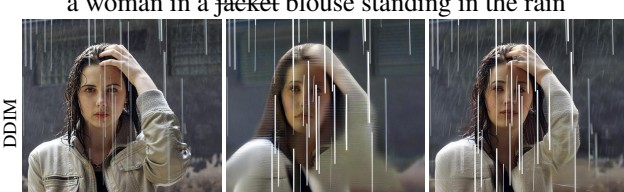

"a ~~gray~~ white horse in the field"

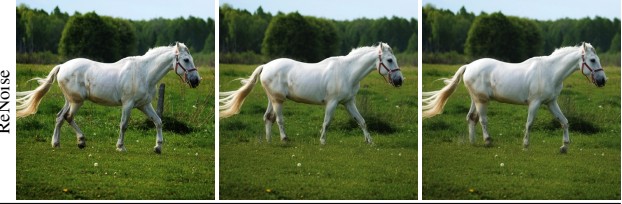

"a golden retriever ~~holding a flower~~ sitting ..."

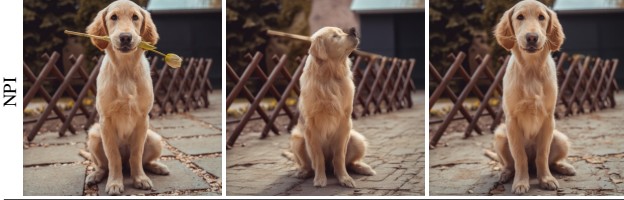

"~~black~~ blue chair in a conference room"

*(a)* Input      *(b)* Baseline      *(c)* w/ Ours

*Figure 4.* Qualitative comparison of image editing results before and after applying our method across different diffusion inversion baselines on SDXL using a total of 50 inversion steps.

**Diffusion Models.** To demonstrate the versatility of our approach, we apply it to a diverse set of diffusion models, including Stable Diffusion v1.5 (SD v1.5) (Rombach et al., 2022), SDXL (Podell et al., 2023), and the more recent few-step model SDXL Turbo (Sauer et al., 2024).

**Implementation.** The experiments are conducted using PyTorch and run on an Nvidia A100. The numbers of inversion and denoising steps for SD (Rombach et al., 2022) and SDXL (Podell et al., 2023) are 50, while for SDXL Turbo (Sauer et al., 2024) they are 4.

**Metrics.** Following established practices in image inversion (Mokady et al., 2023; Alaluf et al., 2021; Ju et al., 2024), we conduct a quantitative evaluation using standard metrics, including Mean Squared Error (MSE), Peak Signal-to-Noise Ratio (PSNR), Learned Perceptual Image Patch Similarity (LPIPS) (Zhang et al., 2018), Structure Similarity Index Measure (SSIM) (Wang et al., 2004), structure distance (Tumanyan et al., 2022), and CLIP similarity (Radford et al., 2021; Wu et al., 2021).

"white plate with ~~fruits~~ pizza on it"

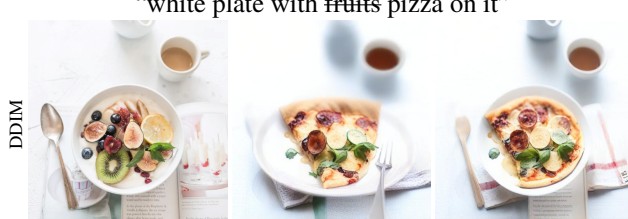

"a cup of coffee with drawing of ~~tulip~~ lion ..."

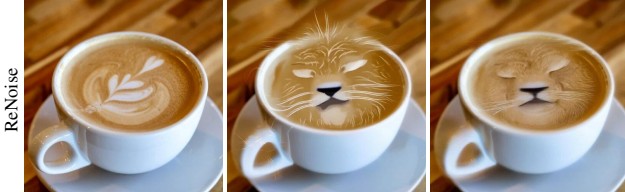

"a painting of a cup with a ~~smoke~~ flower coming out of it"

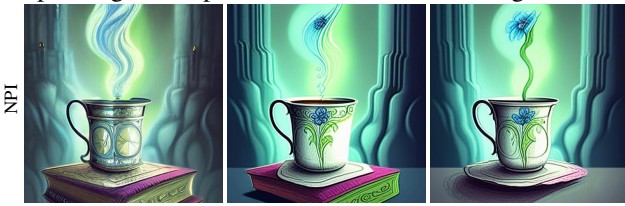

"a ~~colorful~~ red bird standing on a branch"

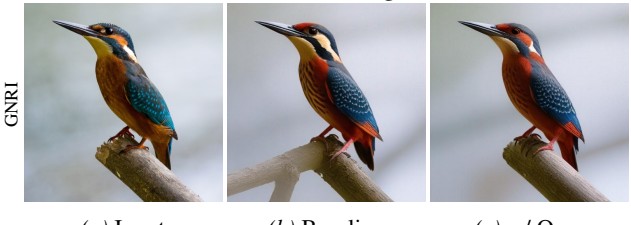

*(a)* Input      *(b)* Baseline      *(c)* w/ Ours

*Figure 5.* Qualitative comparison of image editing results before and after applying our method across different diffusion inversion baselines on SDXL Turbo using a total of 4 inversion steps.

More details in the experimental settings are in Sec. D.

### 4.2. Reconstruction

We evaluate diffusion inversion for image reconstruction on MSCOCO (Lin et al., 2014). Quantitative results on SD v1.5 are reported in Tab. 1. Our timestep rescheduling consistently improves all four baselines without additional computation. Fig. 3 shows qualitative examples where baselines mis-reconstruct the train size/window, fruit shape, and table structure, while our method corrects these errors. Results on SDXL (Podell et al., 2023) and SDXL Turbo (Sauer et al., 2024) are in Sec. D.1.

### 4.3. Editability

We conduct image editing experiments on PIE-Bench (Ju et al., 2024) using four baseline methods with two diffusion models, SDXL (Podell et al., 2023) and SDXL Turbo (Sauer et al., 2024). The quantitative results are presented in Tab. 2.

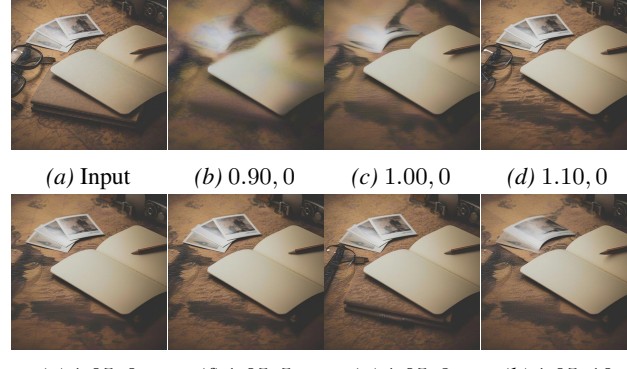

*(a)* Input    *(b)* $0.90, 0$    *(c)* $1.00, 0$    *(d)* $1.10, 0$

*(e)* $1.05, 0$    *(f)* $1.05, 5$    *(g)* $1.05, 8$    *(h)* $1.05, 10$

*Figure 6.* The visual comparison across different settings of the ablated parameter pair $\gamma$ and $d$. The configuration $(1.00, 0)$ corresponds to the conventional DDIM inversion (Song et al., 2021). The editing prompt is to replace "on a map" with "on a carpet".

As shown, our method consistently and in some cases significantly improves structure distance and background preservation while maintaining comparable CLIP similarity with the baselines. It is important to note that few-step models, such as SDXL Turbo (Sauer et al., 2024), experience more severe degradation in background reconstruction. Our method, however, achieves a notable improvement in background preservation, partially addressing this issue.

We also provide visual comparisons on SDXL (Podell et al., 2023) and SDXL Turbo (Sauer et al., 2024) in Figs. 4 and 5. Our method better preserves key details (e.g., faces and object parts) and improves edit fidelity over the baselines. More results are included in Sec. E.

### 4.4. Ablation Studies

In our timestep rescheduling framework, two hyperparameters are tuned: scaling factor $\gamma$ and dynamic-programming window width $d$. We ablate on SDXL (Podell et al., 2023) with DDIM inversion (Song et al., 2021) to assess sensitivity and find a setting balancing background reconstruction and semantic editability. Ablations on SDXL Turbo (Podell et al., 2023) with step number $K = 4$ are in Sec. D.3.

**Scaling factor** $\gamma$ controls global timestep stretching. We vary $\gamma$ with the local adjustment fixed at $d = 0$. Results are in the upper half of Tab. 3. When $\gamma > 1$, background and structure improve, while $\gamma < 1$ degrades some metrics. At $\gamma = 1.10$, LPIPS/SSIM and CLIP similarity start to drop. Thus, we set $\gamma = 1.05$ by default. This matches Sec. 3.3: the early-stage large-error region at small timestep indices is key for fidelity.

**Local window width** $d$ determines the range of local rescheduling. For SDXL (Podell et al., 2023), the default gap is $\Delta t = 20$, so we vary $d$ from 0 to 10 (half the gap). Results are in the lower half of Tab. 3. As $d$ increases, structure retention and background preservation improve. However, at $d = 10$, CLIP similarity decreases, indicating

over-adjustment. We therefore set $d = 8$ by default for balanced performance.

We also show visual comparisons across $\gamma$ and $d$ in Fig. 6. Only (1.05, 8) accurately reconstructs the brown book under the notebook, with color matching the map and carpet.

### 4.5. Runtime for Dynamic Programming

The runtime of our timestep rescheduling is negligible in practice, since the dynamic programming (DP) procedure can be executed offline in advance. As a result, the inversion stage incurs almost no additional runtime overhead. With window width $d = 10$, the DP solver takes 9.32 seconds to compute a schedule for $T = 50$ timesteps on an Intel Xeon w9 CPU, which is negligible compared to the nearly 3 hours required to process the full image reconstruction dataset.

## 5. Conclusion

We study how diffusion timestep choices affect inversion fidelity, an aspect overlooked in prior work of diffusion inversion. We analyze inversion error as a function of step size and timestep index, showing a parabolic pattern with larger deviations at early and late steps. Motivated by this, we propose a nonuniform scheduler that combines global rescaling with local dynamic-programming rescheduling to better allocate compute and reduce total inversion error. The approach is orthogonal to step-wise inversion refinements, improving existing methods without extra parameters or computational overhead. Experiments on image reconstruction and editing confirm consistent gains in visual fidelity and overall performance.

## Acknowledgements

This work was supported by the National Natural Science Foundation of China (No. 62322216, No. U24B20175, No. 62441619, No. 62411540034), Scientific Research Innovation Capability Support Project for Young Faculty (No. SRICSPYF-ZY2025012), and Shenzhen Science and Technology Program (No. KQTD20221101093559018, No. SYSRD20250529113401002).

## Impact Statement

This paper presents work whose goal is to advance the field of machine learning. There are many potential societal consequences of our work, none of which we feel must be specifically highlighted here.

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

# Appendix

In this Appendix, we first provide a complete derivation of Eq. 8 in the main manuscript in Sec. A. Next, we list the existing noise and timestep schedules that are commonly used in Sec. B. We then present a comparison with visual examples of intermediate denoising steps, where the uniformly distributed timestep schedule leads to accumulated inversion errors in Sec. C. Additional experimental results that could not be included in the main manuscript due to page limits are provided in Sec. D, covering ablation studies, image reconstruction, and image editing. Multiple further visual comparisons are presented in Sec. E. We also include a set of video editing results that incorporate our method in Sec. F. Finally, we show a failure case in Sec. G.

## A. Derivation of Eq. 8

Let us begin with DDIM sampling. Given the input latent $\mathbf{z}_t$ and the noise schedule $a_0, a_1, ..., a_t$ with $\alpha_t = \prod_{i=1}^{t} a_i$, the forward diffusion process is expressed as follows,

$$\mathbf{z}_t = \sqrt{\alpha_t}\mathbf{z}_0 + \sqrt{1 - \alpha_t}\boldsymbol{\epsilon}, \ \forall t = \{1, ..., T\} \tag{14}$$

where $\boldsymbol{\epsilon} \sim \mathcal{N}(0, \mathbf{I})$ and $T$ denotes the assigned maximum timestep such that $\mathbf{z}_t$ approximates Gaussian noise. Then the estimated image latent can be written as follows,

$$\hat{\mathbf{z}}_0 = \frac{1}{\sqrt{\alpha_t}}(\mathbf{z}_t - \sqrt{1 - \alpha_t}\boldsymbol{\epsilon}_\theta(\mathbf{z}_t, t, p)), \tag{15}$$

where $\boldsymbol{\epsilon}_\theta(\mathbf{z}_t, t, p)$ estimates the Gaussian noise, parametrized by a neural network with weights $\theta$. Then the forward process is expressed as follows,

$$\mathbf{z}_{t-1} = \frac{\sqrt{\alpha_{t-1}}}{\sqrt{\alpha_t}}(\mathbf{z}_t - \sqrt{1 - \alpha_t}\boldsymbol{\epsilon}_\theta(\mathbf{z}_t, t, p)) + \sqrt{1 - \alpha_{t-1}}\boldsymbol{\epsilon}_\theta(\mathbf{z}_t, t, p). \tag{16}$$

And the reverse process is given by the following expression,

$$\mathbf{z}_t = \frac{\sqrt{\alpha_t}}{\sqrt{\alpha_{t-1}}}(\mathbf{z}_{t-1} - \sqrt{1 - \alpha_{t-1}}\boldsymbol{\epsilon}_\theta(\mathbf{z}_t, t, p)) + \sqrt{1 - \alpha_t}\boldsymbol{\epsilon}_\theta(\mathbf{z}_t, t, p). \tag{17}$$

That is the case of single small step, and we can thus derive the estimation with large step:

$$
\begin{aligned}
\mathbf{z}_t^{(\Delta t)} &= \frac{\sqrt{\alpha_t}}{\sqrt{\alpha_{t-\Delta t}}} \left[ \mathbf{z}_{t-\Delta t} + \sqrt{\alpha_{t-\Delta t}} \left( \sqrt{\frac{1}{\alpha_t} - 1} - \sqrt{\frac{1}{\alpha_{t-\Delta t}} - 1} \right) \boldsymbol{\epsilon}_\theta(\mathbf{z}_t, t, p) \right] \\
&= \frac{\sqrt{\alpha_t}}{\sqrt{\alpha_{t-\Delta t}}} \left[ \mathbf{z}_{t-\Delta t} + \sqrt{\alpha_{t-\Delta t}} \, \Delta\psi(\alpha_t, \Delta t) \, \boldsymbol{\epsilon}_\theta(\mathbf{z}_t, t, p) \right] \\
&= \frac{\sqrt{\alpha_t}}{\sqrt{\alpha_{t-\Delta t}}} \mathbf{z}_{t-\Delta t} + \sqrt{\alpha_t} \, \Delta\psi(\alpha_t, \Delta t) \, \boldsymbol{\epsilon}_\theta(\mathbf{z}_t, t, p).
\end{aligned}
\tag{18}
$$

The additional error caused by the large step can be described by the difference between single-step inversion and the large-step inversion, i.e.,

$$
\begin{aligned}
\delta(\mathbf{z}_t, t, \Delta t) &= \|\mathbf{z}_t^{(\Delta t)} - \mathbf{z}_t^{(1)}\| \\
&= \left\| \frac{\sqrt{\alpha_t}}{\sqrt{\alpha_{t-\Delta t}}} \left[ \mathbf{z}_{t-\Delta t} + \sqrt{\alpha_{t-\Delta t}} \, \Delta\psi(\alpha_t, \Delta t) \, \boldsymbol{\epsilon}_\theta(\mathbf{z}_t, t, p) \right] - \frac{\sqrt{\alpha_t}}{\sqrt{\alpha_{t-1}}} \left[ \mathbf{z}_{t-1} + \sqrt{\alpha_{t-1}} \, \Delta\psi(\alpha_t, 1) \, \boldsymbol{\epsilon}_\theta(\mathbf{z}_t, t, p) \right] \right\| \\
&= \left\| \frac{\sqrt{\alpha_t}}{\sqrt{\alpha_{t-\Delta t}}} \mathbf{z}_{t-\Delta t} + \sqrt{\alpha_t} \, \Delta\psi(\alpha_t, \Delta t) \, \boldsymbol{\epsilon}_\theta(\mathbf{z}_t, t, p) - \frac{\sqrt{\alpha_t}}{\sqrt{\alpha_{t-1}}} \mathbf{z}_{t-1} - \sqrt{\alpha_t} \, \Delta\psi(\alpha_t, 1) \, \boldsymbol{\epsilon}_\theta(\mathbf{z}_t, t, p) \right\|.
\end{aligned}
\tag{19}
$$

*Table 4.* Common noise schedule expressions. The total number of training timesteps used in all schedules is $T = 1000$.

| Noise Schedule | Expression ($i = \frac{t-1}{T-1}$) |
| --- | --- |
| Linear (Ho et al., 2020) | $a_t = 1 - (0.0001 \cdot (1 - i) + 0.02 \cdot i)$ |
| Cosine (Nichol & Dhariwal, 2021) | $a_t = 1 - \min\left(1 - \frac{b_t}{b_{t-1}}, 0.999\right)$, where $b_t = \frac{f(t)}{f(0)}$, $f(t) = \cos\left(\frac{i+0.008}{1+0.008} \cdot \frac{\pi}{2}\right)$ |
| Stable Diffusion (Rombach et al., 2022) | $a_t = 1 - \left(\sqrt{0.00085} \cdot (1 - i) + \sqrt{0.012} \cdot i\right)^2$ |

Given a discrete time interval $\Delta t > 1$, the corresponding large-step diffusion forward process can be formulated as a Gaussian transition $q(\mathbf{z}_{t-1}|\mathbf{z}_{t-\Delta t})$ that moves the latent from $\mathbf{z}_{t-\Delta t}$ to $\mathbf{z}_{t-1}$, and it can be parameterized as:

$$q(\mathbf{z}_{t-1}|\mathbf{z}_{t-\Delta t}) = \mathcal{N}\left(\mathbf{z}_{t-1}; \frac{\sqrt{\alpha_{t-1}}}{\sqrt{\alpha_{t-\Delta t}}}\mathbf{z}_{t-\Delta t}, \left(\sqrt{\alpha_{t-1}}\,\Delta\psi(\alpha_{t-1}, \Delta t-1)\right)^2\mathbf{I}\right), \tag{20}$$

where the mean coefficient $\sqrt{\frac{\alpha_{t-1}}{\alpha_{t-\Delta t}}}$ determines how much signal from $\mathbf{z}_{t-\Delta t}$ is preserved, while the variance term controls the newly injected Gaussian noise at each large step. Therefore, the corresponding sampling form is:

$$\mathbf{z}_{t-1} = \sqrt{\frac{\alpha_{t-1}}{\alpha_{t-\Delta t}}}\,\mathbf{z}_{t-\Delta t} + \sqrt{\alpha_{t-1}}\,\Delta\psi(\alpha_{t-1}, \Delta t-1)\,\boldsymbol{\epsilon}, \qquad \boldsymbol{\epsilon} \sim \mathcal{N}(\mathbf{0}, \mathbf{I}). \tag{21}$$

This equation describes how the latent $\mathbf{z}_{t-\Delta t}$ evolves forward in the diffusion chain by a single large step of size $\Delta t$. For clarity in algorithmic discussions, the above transition can be represented using the shorthand notation:

$$q(\mathbf{z}_{t-1}|\mathbf{z}_{t-\Delta t}) : \mathbf{z}_{t-1} \leftarrow \mathbf{z}_{t-1}^{(\Delta t-1)} = \frac{\sqrt{\alpha_{t-1}}}{\sqrt{\alpha_{t-\Delta t}}}\mathbf{z}_{t-\Delta t} + \sqrt{\alpha_{t-1}}\,\Delta\psi(\alpha_{t-1}, \Delta t - 1)\,\boldsymbol{\epsilon}_\theta(\mathbf{z}_{t-1}, t-1, p), \tag{22}$$

where the arrow indicates an iterative refinement performed within the interval $[t-\Delta t,\ t-1]$. By substituting this expression, $\frac{\sqrt{\alpha_t}}{\sqrt{\alpha_{t-\Delta t}}}\mathbf{z}_{t-\Delta t}$ is canceled out, yielding:

$$\begin{aligned}
\delta &= \|\sqrt{\alpha_t}\,\Delta\psi(\alpha_t, \Delta t)\,\boldsymbol{\epsilon}_\theta(\mathbf{z}_t, t, p) - \sqrt{\alpha_t}\,\Delta\psi(\alpha_{t-1}, \Delta t-1)\,\boldsymbol{\epsilon}_\theta(\mathbf{z}_{t-1}, t-1, p) - \sqrt{\alpha_t}\,\Delta\psi(\alpha_t, 1)\,\boldsymbol{\epsilon}_\theta(\mathbf{z}_t, t, p)\| \\
&= \|\sqrt{\alpha_t}\,\Delta\psi(\alpha_{t-1}, \Delta t-1)\,\boldsymbol{\epsilon}_\theta(\mathbf{z}_t, t, p) - \sqrt{\alpha_t}\,\Delta\psi(\alpha_{t-1}, \Delta t-1)\,\boldsymbol{\epsilon}_\theta(\mathbf{z}_{t-1}, t-1, p)\| \\
&= \|\underbrace{\sqrt{\alpha_t}\,\Delta\psi(\alpha_{t-1}, \Delta t-1)}_{c_{\boldsymbol{\alpha}}(t,\Delta t):\ \text{scaling coefficient}}\underbrace{(\boldsymbol{\epsilon}_\theta(\mathbf{z}_t, t, p) - \boldsymbol{\epsilon}_\theta(\mathbf{z}_{t-1}, t-1, p))}_{\Delta\epsilon_\theta:\text{single-step fixed-point problem}}\|.
\end{aligned} \tag{23}$$

The optimization of $(\boldsymbol{\epsilon}_\theta(\mathbf{z}_t, t, p) - \boldsymbol{\epsilon}_\theta(\mathbf{z}_{t-1}, t-1, p))$ has been examined as a fixed point problem (Garibi et al., 2024; Samuel et al., 2025; Meiri et al., 2023), whereas to the best of our knowledge this work is the first to characterize the inversion error with respect to timesteps in the form $\sqrt{\alpha_t}\,\Delta\psi(\alpha_{t-1}, \Delta t - 1)$.

# B. Discussion of noise and timestep schedules

## B.1. Common Noise Schedule in Diffusion Model

Tab. 4 summarizes the commonly used noise schedule definitions. For all diffusion models evaluated in our experiments, including SD v1.5 (Rombach et al., 2022), SDXL (Podell et al., 2023), and SDXL Turbo (Sauer et al., 2024), the employed noise schedule corresponds to the third case. The table contents are summarized following the formulation provided in (Lin et al., 2024b). We can use this formulation to implement our timestep rescheduling accordingly.

## B.2. Common Timestep Selections in Diffusion Sampling

Tab 5 presents how different sampling implementations select timesteps for few-step inference. The table contents are summarized following the formulation provided in (Lin et al., 2024b). The most widely adopted strategy is the "leading" type used in DDIM (Song et al., 2021) and PNDM (Liu et al., 2022). The table also includes two alternative approaches: "Linspace", which incorporates both the first and last timesteps, and "Trailing", which begins from the final timestep and selects steps backward at a uniform interval. It is evident that all of these timestep schedules rely on uniformly distributed timesteps. Given any such set of timesteps as input, our rescheduling can adaptively reassign them to achieve smaller inversion error.

*Table 5.* Common timestep schedule. The total number of sampling timesteps used during training in all schedules is $T = 1000$ and (K) denotes the number of inference steps.

| Type | Method | Discretization | Timesteps (e.g., $K = 4$) |
|------|--------|----------------|---------------------------|
| Leading | DDIM (Song et al., 2021) | $\text{arange}(1, T + 1, \lfloor T/K \rfloor)$ | 1, 251, 501, 751 |
| Linspace | iDDPM (Nichol & Dhariwal, 2021) | $\text{round}(\text{linspace}(1, T, K))$ | 1, 334, 667, 1000 |
| Trailing | DPM (Lu et al., 2022) | $\text{round}(\text{flip}(\text{arange}(T, 0, -T/K)))$ | 250, 500, 750, 1000 |

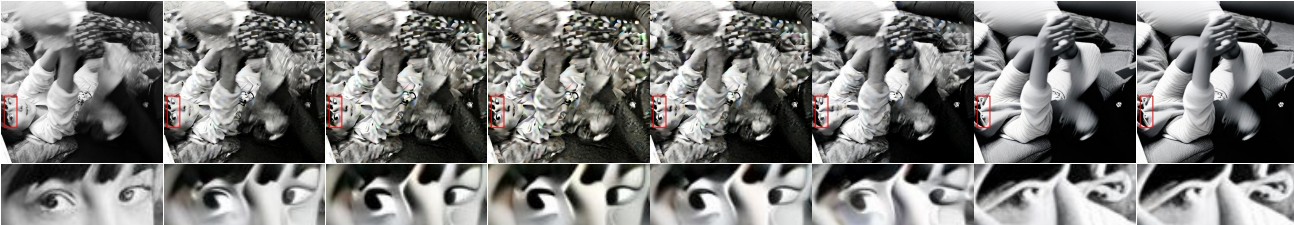

*(a)* Inversion, $t_1$ *(b)* Inversion, $t_2$ *(c)* Inversion, $t_3$ *(d)* Inversion, $t_4$ *(e)* Inference, $t_4$ *(f)* Inference, $t_3$ *(g)* Inference, $t_2$ *(h)* Inference, $t_1$

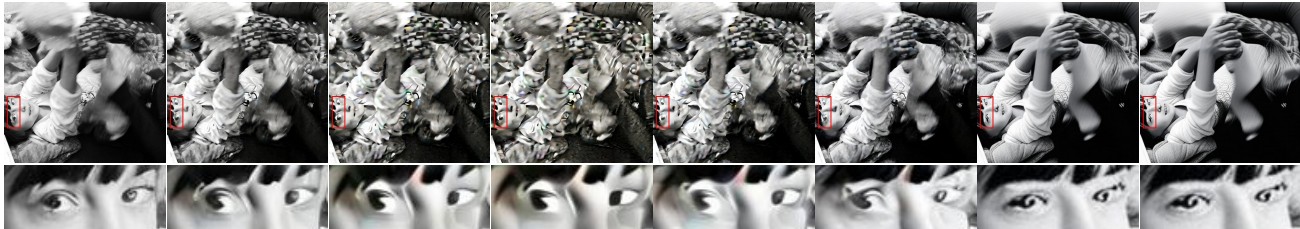

*(i)* Inversion, $t_1$ *(j)* Inversion, $t_2$ *(k)* Inversion, $t_3$ *(l)* Inversion, $t_4$ *(m)* Inference, $t_4$ *(n)* Inference, $t_3$ *(o)* Inference, $t_2$ *(p)* Inference, $t_1$

*Figure 7.* A comparison of the intermediate results obtained via DDIM inversion (Song et al., 2021) with and without our method. Figures (a) to (h) correspond to the conventional DDIM inversion using uniformly distributed timesteps, while the second row shows the results obtained with our timestep rescheduling.

## C. Comparison of Intermediate Results

To provide a clear illustration of how our diffusion inversion timestep rescheduling method improves reconstruction by reducing accumulated inversion errors, we present a comparison of intermediate results during the inversion in Fig. 7. In this case, we use DDIM inversion (Song et al., 2021) with SDXL Turbo (Sauer et al., 2024), which involves only four steps during sampling, making the step-wise visualization clearer. The intermediate visual results are generated by directly decoding the latent representations using the VAE decoder (Kingma & Welling, 2014). The first row shows the intermediate inversion (forward) and inference (backward) results of the diffusion process using uniformly distributed timesteps, while the second row shows the results with our rescheduled timesteps. We observe that conventional DDIM inversion causes severe distortions of the child's double eyelids and eyebrows and incorrectly enlarges the eyes, leading to completely inaccurate final reconstruction. In contrast, our adaptive timestep rescheduling alleviates these issues and preserves the structure of the double eyelids and eyebrows, resulting in a reconstruction that better retains the original facial structure.

## D. More Quantitative Results

### D.1. Image Reconstruction

To comprehensively evaluate the generalizability of our proposed method, we extend our image reconstruction experiments to other diffusion models, including SDXL (Podell et al., 2023) and the fast-sampling SDXL Turbo (Sauer et al., 2024). The experiments are conducted on the MSCOCO (Lin et al., 2014) dataset, consistent with the main manuscript.

**Results on SDXL**. We evaluate our method on the SDXL (Podell et al., 2023) model. The quantitative results are presented in Tab. 6. Our timestep rescheduling improves all inversion baselines in PSNR, SSIM, and LPIPS, confirming its effectiveness on larger and more complex models. Visual examples in Fig. 8 demonstrate that our method better preserves fine details and corrects structural errors produced by the baselines.

*Table 6.* Comparison of image reconstruction performance on the MSCOCO dataset (Lin et al., 2014) using the diffusion model SDXL (Podell et al., 2023). The improvement gains in percentage are highlighted in red.

| Method | PSNR↑ | SSIM$_{\times 10^2}$ ↑ | LPIPS$_{\times 10^3}$ ↓ |
|---|---|---|---|
| DDIM (Song et al., 2021) | 26.68 | 84.61 | 119.18 |
| w/ Ours | 26.89$_{0.79\%}$ | 85.31$_{0.83\%}$ | 103.13$_{13.47\%}$ |
| ReNoise (Garibi et al., 2024) | 28.12 | 87.15 | 85.33 |
| w/ Ours | 28.69$_{2.03\%}$ | 88.58$_{1.64\%}$ | 71.19$_{16.57\%}$ |
| NPI (Miyake et al., 2025) | 26.73 | 84.57 | 117.08 |
| w/ Ours | 26.99$_{0.97\%}$ | 85.13$_{0.66\%}$ | 112.65$_{3.78\%}$ |
| GNRI (Samuel et al., 2025) | 26.79 | 84.81 | 111.42 |
| w/ Ours | 27.20$_{1.53\%}$ | 85.64$_{0.98\%}$ | 103.20$_{7.38\%}$ |

*Table 7.* Comparison of image reconstruction performance on the MSCOCO dataset (Lin et al., 2014) using the diffusion model SDXL Turbo (Sauer et al., 2024). The improvement gains in percentage are highlighted in red.

| Method | PSNR↑ | SSIM$_{\times 10^2}$ ↑ | LPIPS$_{\times 10^3}$ ↓ |
|---|---|---|---|
| DDIM (Song et al., 2021) | 17.09 | 54.55 | 239.36 |
| w/ Ours | 18.48$_{8.13\%}$ | 59.28$_{8.67\%}$ | 207.13$_{13.47\%}$ |
| ReNoise (Garibi et al., 2024) | 20.14 | 65.73 | 176.99 |
| w/ Ours | 21.49$_{6.70\%}$ | 68.61$_{4.38\%}$ | 154.19$_{12.88\%}$ |
| NPI (Miyake et al., 2025) | 18.67 | 58.85 | 201.62 |
| w/ Ours | 19.94$_{6.80\%}$ | 62.92$_{6.92\%}$ | 177.29$_{12.07\%}$ |
| GNRI (Samuel et al., 2025) | 19.21 | 60.51 | 188.61 |
| w/ Ours | 20.07$_{4.48\%}$ | 63.40$_{4.78\%}$ | 174.25$_{7.61\%}$ |

**Results on SDXL Turbo**. We further test our method on the efficient SDXL Turbo (Sauer et al., 2024) model. The results in Tab. 7 and Fig. 9 show a similar trend to SDXL (Podell et al., 2023). Despite the adversarial training and single-step sampling capability of the model, our inversion enhancement successfully improves reconstruction accuracy without requiring additional steps, highlighting its adaptability.

These results indicate that the benefits of timestep rescheduling are not limited to a specific model architecture and are effective for the task of image reconstruction across different diffusion models.

### D.2. Image Editing

To further validate the effectiveness of our proposed method, we conduct extended image editing experiments on the Stable Diffusion (SD) v1.5 model. The experiments are conducted on the PIE-Bench (Ju et al., 2024) dataset, consistent with the main manuscript.

**Extended Results on SD v1.5**. We provide a broader evaluation of our method on the SD v1.5 model (Rombach et al., 2022). The supplementary quantitative results are presented in Tab. 8. Our approach improves all inversion baselines in terms of structure preservation and background consistency, confirming its effectiveness for complex editing tasks. Additional visual examples in Fig. 10 demonstrate that our method more reliably executes the edit instruction while preserving the structural integrity of non-target objects and backgrounds where the baselines fail.

These extended results reinforce that the benefits of our timestep rescheduling method are consistent and observable across a wider range of editing prompts and scenarios for the SD v1.5 architecture (Rombach et al., 2022).

### D.3. More Ablation Studies

This section details the ablation studies on SDXL Turbo (Sauer et al., 2024), conducted with $K = 4$ steps to identify the optimal hyperparameters for our timestep rescheduling framework. The full results are reported in Tab. 9.

*Table 8.* Comparison of diffusion inversion methods on the image editing benchmark PIE-Bench (Ju et al., 2024). The diffusion model used is SD v1.5 (Rombach et al., 2022). The improvement gains in percentage are indicated in red.

| Model | Method | Structure Distance$_{\times 10^3}$ ↓ | PSNR ↑ | Background Preservation LPIPS$_{\times 10^3}$ ↓ | MSE$_{\times 10^4}$ ↓ | SSIM$_{\times 10^2}$ ↑ | CLIP Similariy Whole ↑ | Edited ↑ |
|---|---|---|---|---|---|---|---|---|
| SD v1.5 | DDIM | 27.21 | 23.76 | 104.83 | 64.92 | 80.13 | 23.07 | 20.82 |
| | w/ Ours | 26.48$_{2.76\%}$ | 23.88$_{0.50\%}$ | 101.50$_{3.28\%}$ | 63.22$_{2.69\%}$ | 80.37$_{0.30\%}$ | 23.05$_{0.09\%}$ | 20.81$_{0.05\%}$ |
| | ReNoise | 29.86 | 24.78 | 101.19 | 53.22 | 81.34 | 23.75 | 21.08 |
| | w/ Ours | 28.54$_{4.63\%}$ | 24.83$_{0.20\%}$ | 102.29$_{1.08\%}$ | 52.52$_{1.33\%}$ | 81.34$_{0.00\%}$ | 23.80$_{0.21\%}$ | 21.14$_{0.28\%}$ |
| | NPI | 22.21 | 24.59 | 92.67 | 55.95 | 81.21 | 23.57 | 21.07 |
| | w/ Ours | 20.67$_{7.45\%}$ | 24.76$_{0.69\%}$ | 90.59$_{2.30\%}$ | 54.51$_{2.64\%}$ | 81.40$_{0.23\%}$ | 23.61$_{0.17\%}$ | 21.16$_{0.43\%}$ |
| | GNRI | 25.26 | 24.11 | 99.63 | 61.60 | 80.62 | 23.79 | 21.35 |
| | w/ Ours | 23.57$_{7.17\%}$ | 24.28$_{0.70\%}$ | 97.41$_{2.28\%}$ | 59.93$_{2.79\%}$ | 80.82$_{0.25\%}$ | 23.84$_{0.21\%}$ | 21.39$_{0.19\%}$ |

*Table 9.* The ablation study evaluates the global rescaling factor $\gamma$ and the window width $d$ used in dynamic programming. The diffusion model employed is SDXL Turbo (Sauer et al., 2024), with a total of 4 timesteps ($\Delta t = 250$). The input timesteps follow the "leading" scheduling strategy. The baseline inversion method used for comparison is the DDIM inversion (Song et al., 2021). The best and second-best results in both the upper and lower halves of the table are highlighted in **bold** and underlined, respectively. $c$ is the abbreviation of $\sum_{k=1}^{K} c_{\alpha}(\hat{t}_k, \hat{t}_k - \hat{t}_{k-1})$.

| Settings $\gamma$ | $d$ | Structure Distance$_{\times 10^3}$ ↓ | PSNR ↑ | Background Preservation LPIPS$_{\times 10^4}$ ↓ | MSE$_{\times 10^4}$ ↓ | SSIM$_{\times 10^2}$ ↑ | CLIP Similariy Whole ↑ | Edited ↑ | Error $c$ | Timesteps $\{\hat{t}_k\}_{k=1}^{K=4}$ |
|---|---|---|---|---|---|---|---|---|---|---|
| 1.10 | 0 | 86.07 | 18.19 | 182.38 | 203.94 | 66.23 | **25.81** | 22.52 | **1.5911** | 1, 224, 481, 751 |
| 1.05 | 0 | 84.97 | 18.29 | 182.62 | 199.77 | 66.25 | 25.74 | **22.53** | 1.5971 | 1, 237, 490, 751 |
| 1.00 | 0 | 85.56 | 18.36 | 185.10 | 198.04 | 66.07 | 25.64 | 22.40 | 1.6033 | 1, 251, 501, 751 |
| 0.90 | 0 | **80.52** | **18.72** | **177.92** | **183.27** | **66.85** | 25.58 | 22.42 | 1.6124 | 1, 280, 521, 751 |
| 1.05 | 0 | **84.97** | 18.29 | 182.62 | 199.77 | 66.25 | 25.74 | 22.53 | 1.5971 | 1, 237, 490, 751 |
| 1.05 | 10 | 85.76 | 18.22 | 182.12 | 202.74 | 66.27 | **25.82** | **22.59** | 1.5812 | 3, 227, 480, 741 |
| 1.05 | 25 | 86.15 | **18.30** | 178.34 | **198.12** | **66.46** | 25.72 | 22.53 | 1.5466 | 3, 212, 465, 726 |
| 1.05 | 50 | 88.85 | 18.26 | **176.89** | 198.95 | 66.38 | 25.62 | 22.46 | **1.4878** | 3, 287, 440, 701 |
| 0.90 | 0 | **80.52** | **18.72** | 177.92 | 183.27 | 66.85 | 25.58 | 22.42 | 1.6124 | 1, 280, 521, 751 |
| 0.90 | 10 | 81.99 | 18.66 | 179.03 | 185.78 | 66.66 | 25.59 | 22.36 | 1.5977 | 3, 270, 511, 741 |
| 0.90 | 25 | 82.69 | 18.63 | 178.25 | 186.51 | 66.62 | 25.60 | 22.43 | 1.5615 | 3, 305, 496, 726 |
| 0.90 | 50 | 81.91 | 18.69 | **171.26** | **181.89** | **67.05** | **25.65** | **22.56** | **1.4915** | 3, 230, 571, 701 |

**Analysis of the Scaling Factor** $\gamma$. We first investigate the global stretching parameter $\gamma$ with local adjustment disabled ($d = 0$). The performance trend for SDXL Turbo reveals a distinct preference. As shown in the upper section of Tab. 9, decreasing $\gamma$ to 0.90 yields the best performance across nearly all metrics, including Structure Distance, PSNR, LPIPS, MSE, and SSIM. This indicates that for the few-step SDXL Turbo, stretching the timesteps is more beneficial than concentrating them. We therefore identify $\gamma = 0.90$ as the optimal value, as it provides a substantial boost to both structural and background fidelity.

**Analysis of the Local Window Width** $d$. We subsequently ablate the local window width $d$ for the two optional scaling factors, $\gamma = 1.05$ and $\gamma = 0.90$. The results reveal that local rescheduling is highly effective, with the optimal $d$ being dependent on the chosen $\gamma$. For $\gamma = 1.05$, a moderate local search ($d = 25$) provides the best balance, achieving the top scores in background preservation (LPIPS, MSE, SSIM). For the optimal $\gamma = 0.90$, a more aggressive local search ($d = 50$) is most effective, delivering the best results in LPIPS, MSE, SSIM, and CLIP similarity. This demonstrates that our dynamic programming effectively finds superior local timestep configurations, with the combination $\gamma = 0.90, d = 50$ emerging as the overall optimal configuration for SDXL Turbo.

**Exact Error and Timesteps.** In Tab. 10 and Tab. 9, we provide the exact values of the accumulated error and the corresponding timesteps obtained after our rescheduling for the SDXL (Podell et al., 2023) and SDXL Turbo (Sauer et al., 2024) cases. We observe that the final configurations of the two hyperparameters produce smaller errors compared to the original schedule with uniformly distributed step lengths.

*Table 10.* Ablation study of the global rescaling factor $\gamma$ and window width $d$ for SDXL (Podell et al., 2023) with 50 timesteps ($\Delta t = 20$), using DDIM inversion (Song et al., 2021) and "leading" scheduling for original timestep schedule. Results are evaluated against derived accumulated errors, with corresponding timesteps shown. Best and second-best results per section are **bold** and underlined.

| Settings | | Error | Timesteps |
|---|---|---|---|
| $\gamma$ | $d$ | $\sum_{k=1}^{K} c_{\boldsymbol{\alpha}}(\hat{t}_k, \hat{t}_k - \hat{t}_{k-1})$ | $\{\hat{t}_k\}_{k=1}^{K=4}$ |
| 1.10 | 0 | **2.9594** | 1, 14, 30, 46, 63, 80, 98, 116, 134, 152, 171, 190, 209, 228, 248, 267, 287, 306, 326, 346, 366, 386, 407, 427, 447, 468, 489, 509, 530, 551, 572, 593, 614, 635, 656, 677, 699, 720, 741, 763, 784, 806, 828, 849, 871, 893, 915, 937, 959, 980 |
| 1.05 | 0 | 2.9614 | 1, 17, 35, 53, 71, 90, 109, 128, 147, 166, 185, 205, 224, 244, 263, 283, 303, 323, 343, 363, 383, 403, 423, 443, 464, 484, 504, 525, 545, 565, 586, 606, 627, 648, 668, 689, 709, 730, 751, 772, 792, 813, 834, 855, 876, 897, 918, 939, 960, 980 |
| 1.00 | 0 | 2.9629 | 1, 21, 41, 61, 81, 101, 121, 141, 161, 181, 201, 221, 241, 261, 281, 301, 321, 341, 361, 381, 401, 421, 441, 461, 481, 501, 521, 541, 561, 581, 601, 621, 641, 661, 681, 701, 721, 741, 761, 781, 801, 821, 841, 861, 881, 901, 921, 941, 961, 981 |
| 0.90 | 0 | 2.9651 | 1, 30, 56, 80, 103, 126, 149, 171, 192, 214, 235, 256, 277, 297, 318, 338, 358, 378, 398, 418, 438, 458, 477, 497, 516, 535, 555, 574, 593, 612, 631, 650, 668, 687, 706, 724, 743, 762, 780, 799, 817, 835, 854, 872, 890, 908, 926, 944, 962, 980 |
| 1.05 | 0 | 2.9614 | 1, 17, 35, 53, 71, 90, 109, 128, 147, 166, 185, 205, 224, 244, 263, 283, 303, 323, 343, 363, 383, 403, 423, 443, 464, 484, 504, 525, 545, 565, 586, 606, 627, 648, 668, 689, 709, 730, 751, 772, 792, 813, 834, 855, 876, 897, 918, 939, 960, 980 |
| 1.05 | 2 | 2.9513 | 3, 15, 33, 51, 69, 92, 107, 130, 145, 168, 183, 207, 222, 246, 261, 285, 301, 325, 341, 365, 381, 405, 421, 445, 462, 486, 502, 527, 543, 567, 584, 608, 625, 650, 666, 691, 707, 732, 749, 774, 790, 811, 836, 853, 878, 895, 920, 937, 962, 978 |
| 1.05 | 5 | 2.9166 | 3, 12, 30, 48, 76, 85, 114, 123, 152, 161, 190, 200, 229, 239, 268, 278, 308, 318, 348, 358, 388, 398, 428, 438, 469, 479, 509, 520, 550, 560, 591, 601, 632, 643, 673, 684, 714, 725, 756, 767, 797, 808, 839, 850, 881, 892, 923, 934, 965, 975 |
| 1.05 | 8 | 2.8655 | 3, 9, 43, 45, 79, 82, 117, 120, 155, 158, 193, 197, 232, 236, 271, 275, 311, 315, 351, 355, 391, 395, 431, 435, 472, 476, 512, 517, 553, 557, 594, 598, 635, 640, 676, 681, 717, 722, 759, 764, 800, 805, 842, 847, 884, 889, 926, 931, 968, 972 |
| 1.05 | 10 | **2.8342** | 3, 7, 25, 43, 78, 80, 116, 118, 154, 156, 193, 195, 234, 236, 271, 273, 313, 315, 351, 353, 393, 395, 431, 433, 474, 476, 513, 515, 553, 555, 594, 596, 637, 639, 677, 679, 718, 720, 761, 763, 801, 803, 843, 845, 885, 887, 928, 930, 968, 970 |

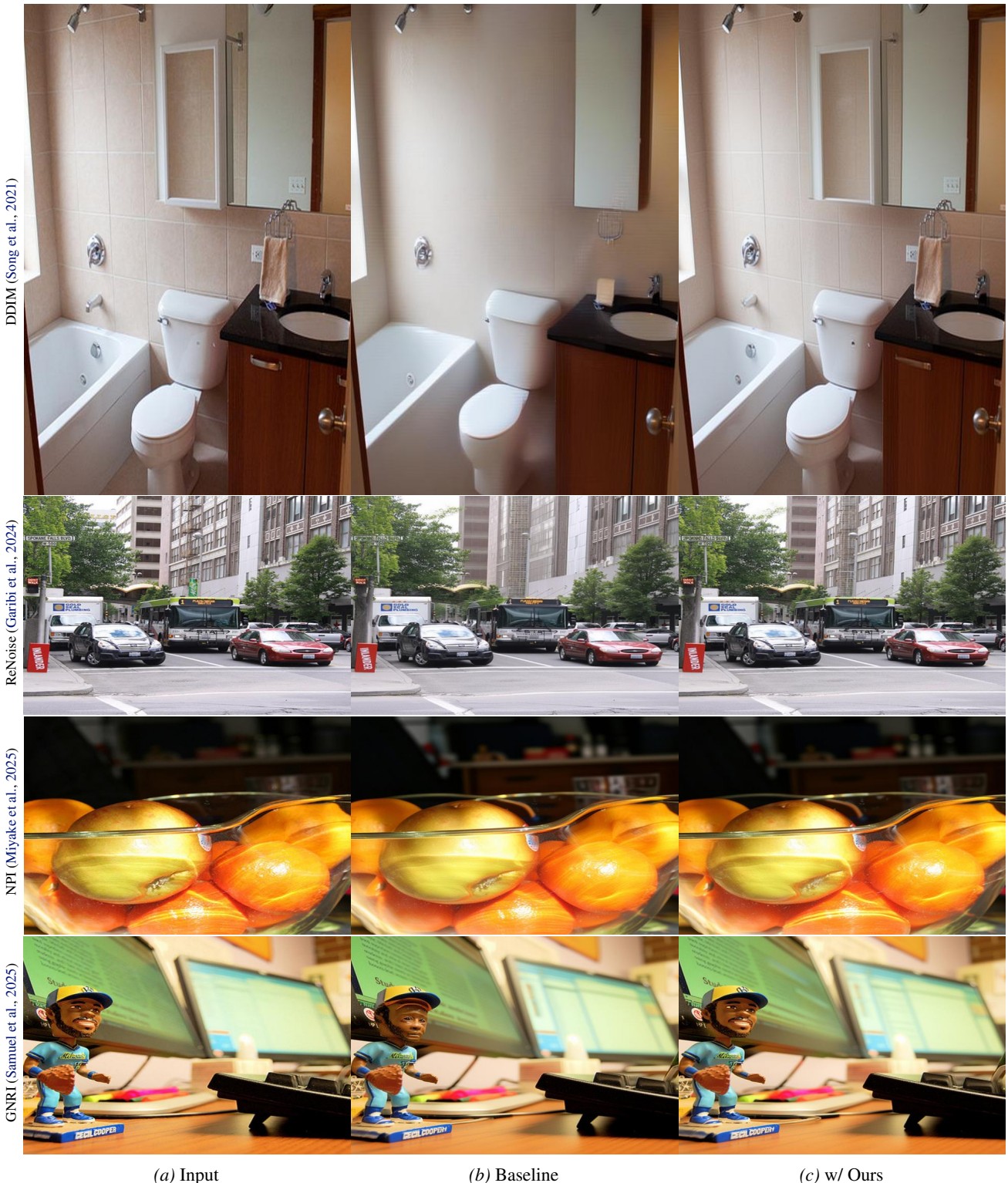

*(a)* Input        *(b)* Baseline        *(c)* w/ Ours

*Figure 8.* Qualitative comparison of image reconstruction results before and after applying our timestep rescheduling method across different diffusion inversion baselines on SDXL (Podell et al., 2023) using a total of 50 steps.

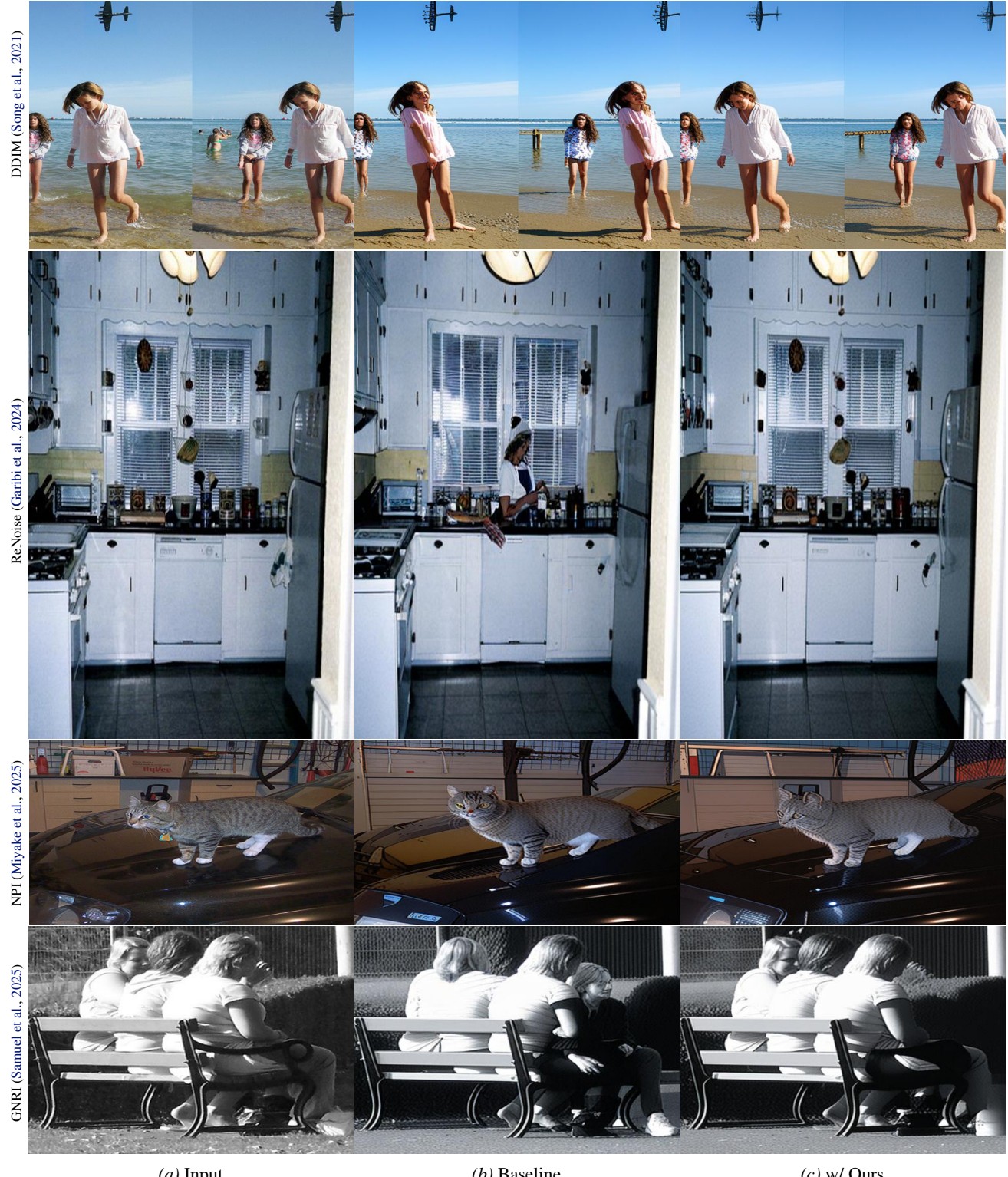

*(a)* Input                     *(b)* Baseline                     *(c)* w/ Ours

*Figure 9.* Qualitative comparison of image reconstruction results before and after applying our timestep rescheduling method across different diffusion inversion baselines on SDXL Turbo (Sauer et al., 2024) using a total of 4 steps.

## E. More Visual Comparisons

### E.1. Image Reconstruction

This section provides supplementary visual evidence for the image reconstruction performance discussed in the main manuscript. We include an expanded set of qualitative results from PIE-Bench (Ju et al., 2024) for both SDXL (Podell et al.,

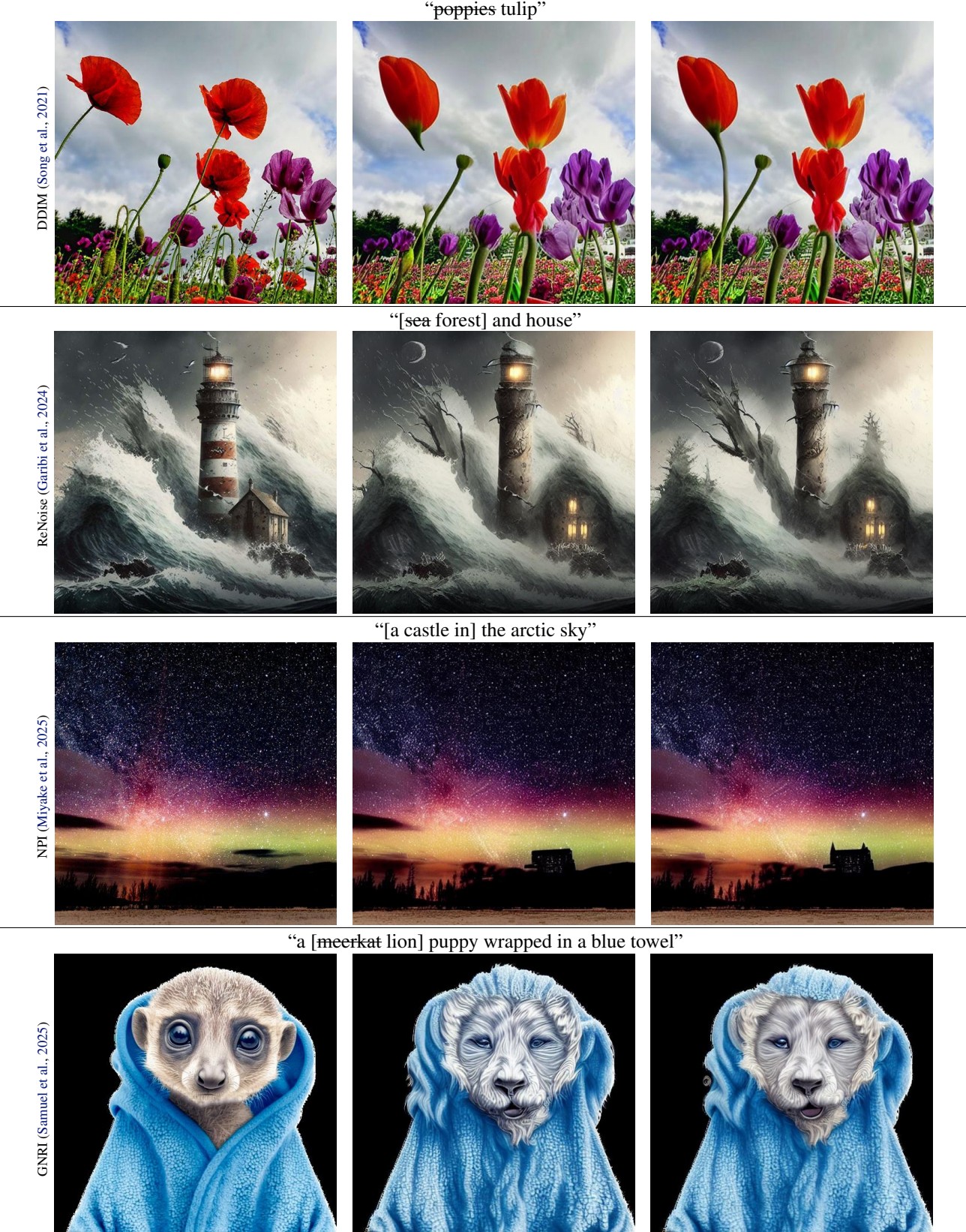

*(a)* Input            *(b)* Baseline            *(c)* w/ Ours

*Figure 10.* Qualitative comparison of image editing results before and after applying our method across different diffusion inversion baselines on SD v1.5 (Rombach et al., 2022) using a total of 50 inversion steps.

2023) and SDXL Turbo (Sauer et al., 2024) models, covering all four baseline inversion methods.

As shown in Fig. 8, applying our method to SDXL demonstrates superior robustness in complex editing scenarios. It more reliably preserves the structure of the washroom in the first case, the details of the background building in the second case, the light and shadow of oranges in the third case, and the facial features of the garage kit in the last case. Our approach maintains the structural integrity of backgrounds and objects while preventing unwanted artifacts in detailed regions where the baseline methods fail.

The challenges are more pronounced for the few-step SDXL Turbo model, as illustrated in Fig. 9. Baseline inversions often produce significant background degradation or reduced edit fidelity. Our method consistently mitigates these issues, enabling SDXL Turbo to perform manipulations with substantially improved background preservation and overall semantic coherence, thereby enhancing the practical utility of fast-sampling models for editing tasks.

### E.2. Image Editing

This section provides supplementary visual evidence for the image editing performance discussed in the main manuscript. We include a broader set of qualitative results from PIE-Bench (Ju et al., 2024) for the SD v1.5 (Rombach et al., 2022) model, covering all four baseline inversion methods.

As illustrated in Fig. 10, our method demonstrates superior robustness in complex editing scenarios. For instance, it generates more natural flower and floral axes in the first case, reduces the appearance of undesired waves in the second case, preserves the mimic shape of the castle in the third case, and minimizes unwanted blue tones in the lion face and fur in the last case. It more reliably preserves fine details and background context, maintains the structural integrity of primary objects, and prevents the introduction of unwanted artifacts where baseline methods fail. The challenges of achieving faithful edits are evident in the baseline results, which frequently lead to partial edit execution or degradation of unrelated image regions. Our method consistently addresses these issues, enabling more precise manipulations with markedly improved semantic coherence and overall edit fidelity.

## F. Video Editing

We also evaluate our method on the task of video editing. We apply our timestep rescheduling to Videoshop (Fan et al., 2024), a recent diffusion inversion-based video editing method. Videoshop (Fan et al., 2024) takes an edited first frame generated by DDIM inversion (Song et al., 2021) and propagates the edit to the subsequent frames. We conduct experiments on two video clips provided by the authors, with the results shown in Fig. 11 and Fig. 12.

In the first case, the editing task is to change the color of a sheep to black. In the baseline results, the subsequent frames gradually deviate from the original shapes and motion of the sheep, whereas our method better preserves the fidelity of the original shape while maintaining the prompted black color.

In the second case, the first frame adds two zebras to the right region of the scene. The baseline progressively removes the zebras in later frames, while our method successfully maintains their presence throughout the video.

These cases demonstrate the generalizability and effectiveness of our timestep rescheduling for video-based diffusion inversion.

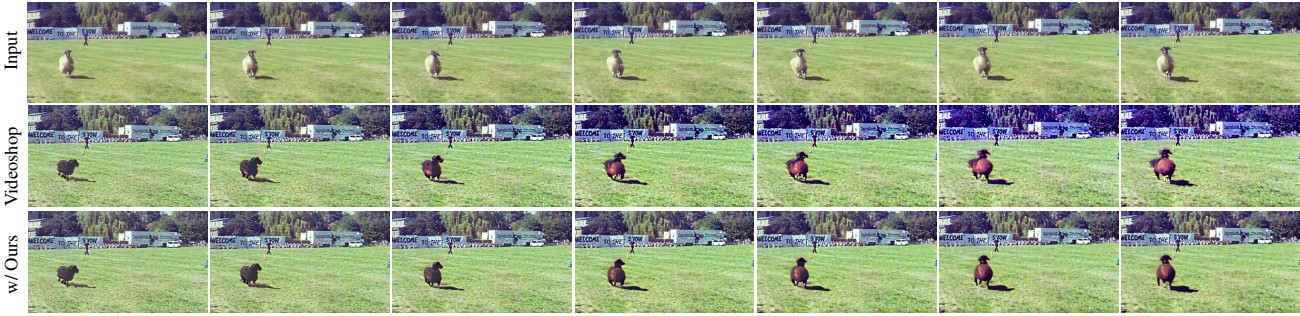

*Figure 11.* A visual example of inversion-based video editing before and after integrating our timestep rescheduling in Videoshop (Fan et al., 2024). The rows from top to bottom are the frames of input video, the result of Videoshop (Fan et al., 2024) and the result with ours.

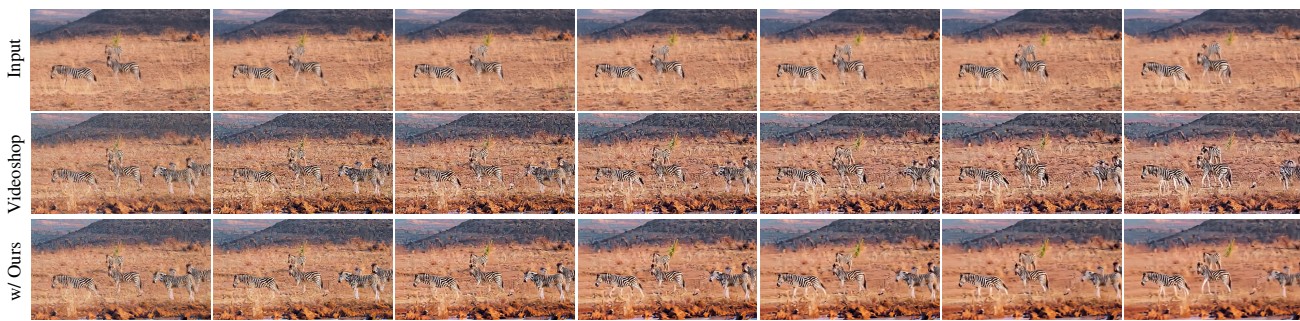

*Figure 12.* A visual example of inversion-based video editing before and after integrating our timestep rescheduling in Videoshop (Fan et al., 2024). The rows from top to bottom are the frames of input video, the result of Videoshop (Fan et al., 2024) and the result with ours.

"a ~~young~~ old woman is holding a dog"

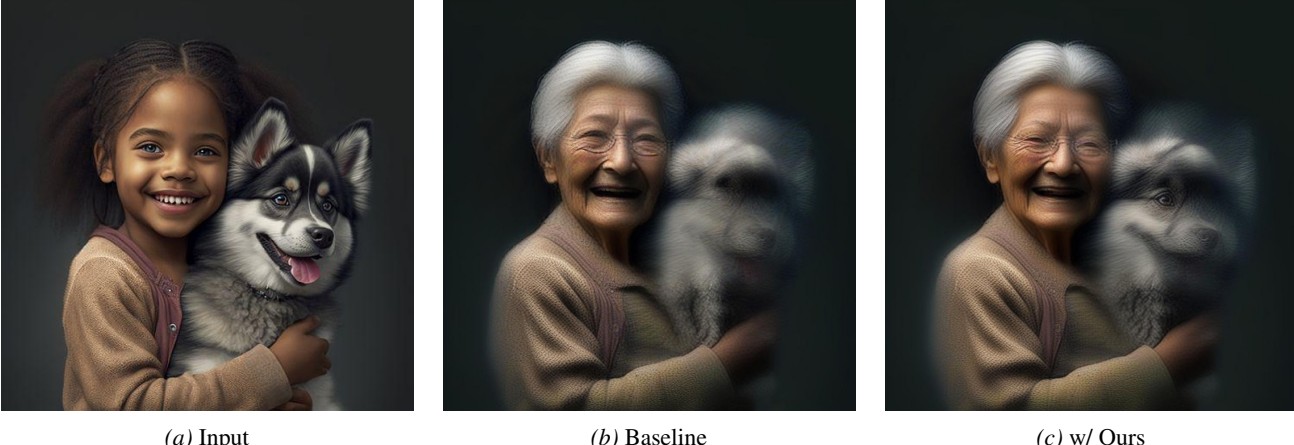

| *(a)* Input | *(b)* Baseline | *(c)* w/ Ours |

*Figure 13.* A failure case of image editing results for GNRI (Samuel et al., 2025) diffusion inversion baseline on SDXL (Podell et al., 2023) using a total of 50 inversion steps.

## G. Failure Cases

We illustrate a representative failure case in Fig. 13, where both the baseline and our method struggle to preserve the dog's features while editing the woman's face. Notably, our method demonstrates superior structural consistency by successfully reconstructing at least one dog's eye, whereas the baseline fails entirely.

