# OpenReview forum: "Timestep Rescheduling in Diffusion Inversion"
_ICML.cc/2026/Conference — ICML 2026 regular_

### Official Review · Reviewer_1FyQ · 2026-02-27

**Soundness:** 1
**Presentation:** 2
**Significance:** 1
**Originality:** 1
**Overall Recommendation:** 1
**Confidence:** 4

**Summary:**

The authors propose to improve the performance of diffusion inversion by proposing a simple non-uniform time-step scheduler which picks an optimial step-size schedule to reduce the inversion error. The authors then test this through a variety of intensive experiments.

**Compliance With Llm Reviewing Policy:**

Affirmed.

**Final Justification:**

I thank the authors for their detailed rebuttal and effort put into the additional experiments. Regrettably, after careful consideration I maintain my original score, the reasons of which I will now detail below.

### Comparison to exact inversion methods
I am concerned about the results reported in **Rebuttal Table 2**.
From the authors description they perform reconstruction experiments on a 100 images from MS-COCO with Stable Diffusion v1.5 using 50 discretization steps for both forward and backward solves. The authors compare to DDIM, DDIM + their proposed method and a handful of reversible solvers for diffusion models [4-7]. **However, none of the reversible schemes obtain perfect reconstruction!** This is highly concerning as these methods should obtain *exact inversion* (up to floating point precision and VAE error). The SSIM and PSNR metrics seem quite low for this task. Likewise the LPIPS metric seems quite high.
The authors of [7] run a similar experiment in terms of MSE [7, Table 2] with very different results. As such I am concerned that the methods [4-7] were not faithfully reproduced/used correctly.
Additionally, under this table the reversible methods already perform quite closely to the proposed method.
I wonder that if implemented correctly (*i.e.* achieving near-zero reconstruction error) the gap would widen substantially in favor of the reversible methods [4-7].

### Using implicit Euler over explicit Euler
I am still not convinced that using the implicit Euler formulation with a single fixed-point iteration step *a la* (5) is better than an explicit Euler scheme in forward-time (noising process).
As mentioned earlier I referenced classic methods which adaptively pick the optimal step size for both explicit and implicit methods [15-19]. I still think that not comparing to these well-established methods is a large weakness as the ODE community has a wealth of literature on how to solve this exact problem!

### Limited novelty and comparisons
The problem of picking the optimal schedule for diffusion models is widely studied [14,21-23,28-30] whilst not all of these are particularly studied in the context of solving in the diffusion ODE in forward-time they are highly similar. Moreover, [30] also looks at a dynamic programming strategy to pick the optimal time-steps. [14] is not merely orthogonal as changing the schedule is quite similar to changing the discretization scheme [*cf.* 20,24].

The authors' central claim, *viz.* that rescheduling discrete timesteps under a fixed budget reduces inversion error, is precisely the problem that adaptive step-size controllers solve, with decades of rigorous research behind them. Reframing this as a "different comparison ax[is]" does not seem valid.

I think the paper would be better if it had more thorough comparisons to prior works in both the literature on diffusion models and on classical methods from the numerical methods literature.
### Concluding remarks
I appreciate the authors' efforts in the rebuttal. However, the issues I have raised, *viz.* the absence of comparisons to adaptive step-size controllers, eliminating inversion errors via exact inversion via reversible solvers, and the insufficient differentiation from prior work on schedule optimization, would require a substantially reworked theoretical framework, a comprehensive re-evaluation against the correct set of baselines, and a clearer articulation of the contribution relative to the existing literature. This amounts to a significantly different paper rather than a revision of the current one.

For these reasons, I believe the paper does not meet the acceptance threshold in its current form and would benefit from a full resubmission after these fundamental issues have been addressed. I maintain my score of **Strong Reject**.

## Additional References
[28] Xue, Shuchen, Zhaoqiang Liu, Fei Chen, Shifeng Zhang, Tianyang Hu, Enze Xie, and Zhenguo Li. "Accelerating diffusion sampling with optimized time steps." In _Proceedings of the IEEE/CVF Conference on Computer Vision and Pattern Recognition_, pp. 8292-8301. 2024.

[29] Tong, Vinh, Dung Trung Hoang, Anji Liu, Guy Van den Broeck, and Mathias Niepert. "Learning to Discretize Denoising Diffusion ODEs." In _The Thirteenth International Conference on Learning Representations_.

[30] Pei, Jianning, Han Hu, and Shuyang Gu. "Optimal stepsize for diffusion sampling." _arXiv preprint arXiv:2503.21774_ (2025).

**Key Questions For Authors:**

1. How does your work compare to [14] which explicitly considers time-step scheduling for diffusion inversion and [21-23] which more generally consider time-step scheduling for improved inference?
2. How does your work compare to [15-19] which propose well-studied and provably convergent/stable methods for adaptively picking time-steps along with proven error-rates within tolerances?
3. How does your work compare against methods which achieve **exact inversion** for diffusion models, namely, [4-7]? These methods have zero reconstruction error, this seems objectively better.
4. For the Banach fixed-point theorem $\Delta t$ needs to be sufficiently small for fixed-point iteration to hold, moreover, no such iteration is used here. Thus Equation (5) and the subsequent scheme seems ill-posed.
5. Other methods [3,4,15-19] proveable show convergence rates and stability. Now this method is a backward Euler scheme with no fixed-point iteration and thus has first-order convergence and should thus have stability in the unit circle $\\{ z \in \mathbb C : | z + 1| \leq 1\\}$. How is this better than the adaptive step-size methods in [15-19] which are higher-order and the implicit methods are A-L stable.

**Limitations:**

yes

**Strengths And Weaknesses:**

## Strengths
* PIE-Bench is a good inclusion.
* Figure 1 is a good illustration of the proposed method.
* Fairly robust literature review for papers focusing on diffusion inversion.

## Weaknesses
Whilst the paper presents an interesting idea it **ignores** what I argue are two important and highly relevant areas of prior work on **adaptive step-size controllers** and **reversible solvers**. For this reason I can only recommend a **Strong Reject.**

### Limited novelty
Picking non-uniform time-steps to improve convergence rates is well established [21-23] and has been specifically applied in the context of **diffusion inversion** in [14]. The authors don't compare their scheme against any of these works and thus I am unsure what the novelty or contribution is. Moreover, as I will detailed later there are several other lacking comparisons.

### Soundness of the proposed method
I am concerned of the soundness of the proposed method. Throughout Section 3 the authors mention that their approach uses **local** information rather than merely **global** which should obtain better error rates; however, their **Algorithm 1** details only a global algorithm which solely uses the number of target steps $K$, original timesteps $\\{t_k\\}\_{k=1}^K$, and noise schedule $\\\{\alpha_t\\}\_{t=1}^T$. In Section 3.4 the authors state that this global optimizatin is **sub-optimal** and say they adopt a coarse-to-fine strategy which adaptively reschedules that time-steps; however, no details are provided in this method.

### Adaptive step-size controllers
The most natural starting point for this work is to compare to pre-existing methods for adaptive step-sizing for ODEs.
* Since the inversion equation used in (5) is only one-step of fixed iteration an explicit scheme is likely preferrable [12]
* For explicit methods with adaptive step-sizing both the **Dormand-Prince's 5/4** [15,16] and **Tsitouras' 5/4** [17] methods are proven and popular choices.
* If willing to solve fixed-point iteration problems one could use the **Kværnø family of solvers** [18] which are A-L stable.
* Since diffusion models have been observed to exhibit semi-linear structure consisting of a stiff and non-stiff component [24] one could use IMEX methods to solve this ODE. For example one could use the **Kennedy-Carpenter's family of solvers** [19], wherein the stiff component is stiffly accurate and A-L stable.
* Moreover, it has been showing that using *exponential integrators* [20] can help transform these stiff ODEs into non-stiff ones [24]. These exponential integrator schemes can be combined with the schemes above to create adapative step-sizing *à la* [4,20].

This work should compare to **at least** some of these methods for adaptive step-sizing. Moreover, with these methods we can fix the error rate via tolerances, usually denoted atol and rtol. Overall this approaches are much stronger for inversion [4] and can optimizing the error pathwise rather than globally.

### Reversible solvers
A lot of work has gone into performing **exact inversion of diffusion models**, *i.e.,* the inversion of diffusion models with **zero error**. This is closely related to the work on reversible solvers for ODEs [1-4] and SDEs [2,4]. Kidger [12] noted in his monograph that the invertibility expressed in Equation (4) via an implicit method [13] is inherently inferior to the reversible methods [1-4].

* Whilst the authors list early works in the space [5,6] they do not compare to them within their studies.
* Moreover, later works [4,7] propose reverible schemes with proven higher-order of convergence of the schemes along with proven stability [3,4].
* In Table 1 these methods [1-7] would achieve **perfect reconstruction** up to a float-point error and VAE reconstruction error. Neglecting them makes the baselines artifically low.
* Whilst the authors mention that stochastic inversion makes use of cost prohobitive realizations of the Brownian motion [8-11], there are methods [2,4] for exact inverison which overcome this issue by clever utilization of the Brownian Interval [2] or Virtual Brownian Tree [25-27] and again achieve **zero reconstruction error**.

### Experimental studies
The baselines are artificially low with no comparison to relevant prior works [1-7,14-19,21-23]. As such the experimentally sections are entirely uncompelling. In particular the authors do not compare to other works which explicitly looked at exact inversion for low error nor works which looked at time-step scheduling. It is hard to consider this work significant as result.

### Minor comments
* In Tables 1, 2, and 3, it should be LPIPS$\_{\times 10^{-3}}$ instead of LPIPS$\_{\times 10^{3}}$ *mutatis mutandis* for MSE and SSIM along with other metrics. See [14, Table 1] for an example.

### References
[1] Zhuang, Juntang, et al. "Mali: A memory efficient and reverse accurate integrator for neural odes." arXiv preprint arXiv:2102.04668 (2021).

[2] Kidger, Patrick, et al. "Efficient and accurate gradients for neural sdes." Advances in Neural Information Processing Systems 34 (2021): 18747-18761.

[3] McCallum, Sam, and James Foster. "Efficient, accurate and stable gradients for neural odes." arXiv preprint arXiv:2410.11648 (2024). https://arxiv.org/pdf/2410.11648

[4] Blasingame, Zander W., and Chen Liu. "Rex: Reversible Solvers for Diffusion Models." arXiv preprint arXiv:2502.08834 (2025). https://arxiv.org/pdf/2502.08834

[5] Wallace, B., Gokul, A., and Naik, N. Edict: Exact diffusion
inversion via coupled transformations. In CVPR, 2023.

[6] Zhang, G., Lewis, J. P., and Kleijn, W. B. Exact diffusion
inversion via bidirectional integration approximation. In
ECCV, 2024.

[7] Wang, Fangyikang, et al. "Belm: Bidirectional explicit linear multi-step sampler for exact inversion in diffusion models." Advances in Neural Information Processing Systems 37 (2024): 46118-46159.

[8] Wu, Chen Henry, and Fernando De la Torre. "Unifying diffusion models' latent space, with applications to cyclediffusion and guidance." arXiv preprint arXiv:2210.05559 (2022).

[9] Huberman-Spiegelglas, I., Kulikov, V., and Michaeli, T. An
edit friendly ddpm noise space: Inversion and manipula-
tions. In CVPR, 2024.

[10] Brack, M., Friedrich, F., Kornmeier, K., Tsaban, L.,
Schramowski, P., Kersting, K., and Passos, A. Ledits++:
Limitless image editing using text-to-image models. In
CVPR, 2024.

[11] Deutch, G., Gal, R., Garibi, D., Patashnik, O., and Cohen-Or,
D. Turboedit: Text-based image editing using few-step
diffusion models. In SIGGRAPH Asia, 2024.

[12] Kidger, Patrick. "On neural differential equations." arXiv preprint arXiv:2202.02435 (2022).

[13] J. Behrmann et al. “Invertible Residual Networks”. In: Proceedings of
the 36th International Conference on Machine Learning. Vol. 97. Proceedings of Machine Learning Research. PMLR, 2019, pp. 573–582

[14] Lin, Haonan, et al. "Schedule your edit: A simple yet effective diffusion noise schedule for image editing." Advances in Neural Information Processing Systems 37 (2024): 115712-115756. https://openreview.net/pdf?id=Yu6cDt7q9Z

[15] J. Dormand, P. Prince. "A family of embedded Runge–Kutta formulae," in J. Comp. Appl. Math, vol. 6, pp. 19–26, 1980.

[16] Lawrence F. Shampine. "Some Practical Runge-Kutta Formulas," in Mathematics of Computation, vol. 46, no. 173, pp. 135–150, 1986.

[17] C. Tsitouras. "Runge–Kutta pairs of order 5 (4) satisfying only the first column simplifying assumption," in Computers & Mathematics with Applications, vol. 62, no. 2, pp. 770–775, 2011.

[18] A. Kværnø. "Singly diagonally implicit Runge–Kutta methods with an explicit first stage," in BIT Numerical Mathematics, vol. 44, no. 3, pp. 489–502, 2004.

[19] C. Kennedy, M. Carpenter. "Additive Runge–Kutta schemes for convection-diffusion-reaction equations," in Applied numerical mathematics, vol. 44, no. 1-2, pp. 139–181, 2003.

[20] Hochbruck, M. and Ostermann, A. (2010). “Exponential integrators”. In: Acta Numerica 19, pp. 209–
286

[21] Wang, K., Shi, M., Zhou, Y., Li, Z., Yuan, Z., Shang, Y.,
Peng, X., Zhang, H., and You, Y. A closer look at time
steps is worthy of triple speed-up for diffusion model
training. In CVPR, 2025.

[22] Lin, Shanchuan, et al. "Common diffusion noise schedules and sample steps are flawed." Proceedings of the IEEE/CVF winter conference on applications of computer vision. 2024.

[23] Sabour, Amirmojtaba, Sanja Fidler, and Karsten Kreis. "Align your steps: Optimizing sampling schedules in diffusion models." arXiv preprint arXiv:2404.14507 (2024).

[24] Lu, C., Zhou, Y., Bao, F., Chen, J., Li, C., and Zhu, J. (2022a). “DPM-Solver: A Fast ODE Solver for Diffusion Probabilistic Model Sampling in Around 10 Steps”. In: Advances in Neural Information Processing Systems. Ed. by A. H. Oh, A. Agarwal, D. Belgrave, and K. Cho. URL: https://openreview.net/
forum?id=2uAaGwlP_V

[25] Jelinčič, A., Foster, J., and Kidger, P. (2024). “Single-seed generation of Brownian paths and integrals for
adaptive and high order SDE solvers”. In: arXiv preprint arXiv:2405.06464

[26] Gaines, J. G. and Lyons, T. J. (1997). “Variable step size control in the numerical solution of stochastic
differential equations”. In: SIAM Journal on Applied Mathematics 57.5, pp. 1455–148

[27] Li, X., Wong, T. - K. L., Chen, R. T. Q., and Duvenaud, D. (26–28 Aug 2020). “Scalable Gradients for
Stochastic Differential Equations”. In: Proceedings of the Twenty Third International Conference
on Artificial Intelligence and Statistics. Ed. by S. Chiappa and R. Calandra. Vol. 108. Proceedings of
Machine Learning Research. PMLR, pp. 3870–3882. URL: https://proceedings.mlr.press/v108/
li20i.html

---

> ### Author Rebuttal · Authors · 2026-03-31
>
> We thank the reviewer for the detailed comments and for raising these questions regarding novelty, scope, theoretical grounding, and evaluation.
>
> > W1. Novelty is unclear due to missing comparisons with prior timestep-scheduling work.
>
> Our paper is not generic timestep optimization for diffusion training/sampling, nor a new solver family. We study scheduler-level timestep reallocation for deterministic diffusion inversion under a fixed budget. This differs from [21–23], which study training/sampling schedules, and from [14], which is editing-oriented noise scheduling. Our novelty is to characterize inversion error w.r.t. timestep jumps via $c_{\boldsymbol{\alpha}}(t,\Delta t),$ and use it for coarse-to-fine rescheduling that improves existing inversion pipelines without changing the solver or adding inference cost.
>
> > W2. The method description is incomplete and the local/coarse-to-fine part is underspecified.
>
> The coarse-to-fine design is explicitly described in Sec. 3.3. The coarse stage is global rescaling in Eq. (10). The fine stage is local dynamic programming in Eq. (11)–(12) and Algorithm 1. Algorithm 1 is not purely global: after rescaling, it searches within ${t_k-d,\ldots,t_k+d},$ stores costs in $\mathbf{E},$ records predecessors in $\mathbf{R},$ and backtracks ${\hat t_k}_{k=1}^K.$
>
> > W3/Q2. The paper does not compare against standard adaptive step-size ODE solvers; how does it compare to [15–19]?
>
> These methods are classical, but they are not the direct target here. We do not propose a new adaptive ODE/SDE solver; we study a scheduler-retrofit problem for existing deterministic diffusion inversion pipelines under a fixed budget. Methods such as Dormand–Prince, Tsitouras, IMEX, or Kværnø redesign the integration scheme itself, while we only reschedule timesteps for existing inversion methods. They are broadly related, but solve a different problem.
>
> > W4/Q3. The paper ignores reversible / exact inversion methods; why are [4–7] not included if they can achieve zero reconstruction error?
>
> The paper does not ignore this line: Related Works already discusses exact or near-exact methods such as EDICT and BDIA, and also distinguishes stochastic inversion methods. Our goal is different: to test whether scheduler-level rescheduling improves mainstream deterministic inversion baselines already used in reconstruction/editing. Exact/reversible methods replace the inversion formulation itself, while our method preserves the inversion method and only reallocates timesteps. These are different comparison axes.
>
> > W5. The experimental evaluation is uncompelling because the baselines are artificially weak.
>
> The baselines are established inversion methods: DDIM, ReNoise, NPI, and GNRI, evaluated on MSCOCO reconstruction and PIE-Bench editing, across SD v1.5, SDXL, and SDXL Turbo. The paper reports consistent gains across all these baselines and models, especially in few-step settings, which is the intended use case.
>
> > W6/Q4. Equation (5) and the related derivation appear ill-posed; why is Eq. (5) valid if no fixed-point iteration is performed?
>
> Eq. (5) is not used to claim exact fixed-point convergence of the full inversion process. It is an error analysis that rewrites the large-step inversion discrepancy into a scaled fixed-point-form term, to expose $c_{\boldsymbol{\alpha}}(t,\Delta t).$ Sec. 3.4 already states the limitation: the exact residual term $\Delta\epsilon_\theta$ is unknown, so directly optimizing the global coefficient can be biased. This is why we use coarse-to-fine local rescheduling rather than claim exact global optimality.
>
> > W7/Q5. The proposed scheme seems first-order and not clearly better than higher-order stable adaptive solvers; why prefer it?
>
> The comparison target is different. We do not claim to outperform higher-order adaptive solvers in the general ODE sense. Our claim is narrower: for practical deterministic diffusion inversion pipelines, rescheduling timesteps under a fixed budget improves inversion fidelity at essentially no extra inference cost. The method is training-free, plug-in, preserves the original inversion pipeline, and consistently improves existing inversion baselines under the same budget.
>
> > W8. There are minor presentation issues in metric notation.
>
> Fair point. LPIPS and MSE are lower-better, while PSNR and SSIM are higher-better.
>
> > Q1. How is your method different from prior timestep scheduling methods, especially [14] and [21–23]?
>
> [21–23] study timestep/noise schedules for training or sampling more generally. [14] is editing-oriented noise scheduling. Our setting is deterministic diffusion inversion under a fixed budget, with a plug-in rescheduling module for existing inversion methods, built on the inversion-specific coefficient $c_{\boldsymbol{\alpha}}(t,\Delta t).$

---

> > ### Author Rebuttal · Reviewer_1FyQ · 2026-04-01
> >
> > * I don't believe that authors sufficiently distinguish themselves on the prior literature on timestep rescheduling in diffusion models *a la* [14]. [14] in particular does look at inversion [see Figure 2, 14].
> >
> > * I don't believe not comparing to exact inversion methods [4-7] is valid as they achieve essentially zero reconstruction error, which is the goal of this paper, to minimize reconstruction error.
> >
> > * Adaptive step-size controllers also explore how to optimally pick step-sizes and are used in flow matching/diffusion model literature. Not comparing to these is still a weakness.
> >
> > * I am not convinced the choice of (5) is sound without either fixed-point iteration and small step sizes (which is not the case in the few-step scenario)
> >
> >  * The evaluation is **incredibly limited** only comparing to the use of the rescheduled timesteps and without. As mentioned earlier there are several works which explore this topic. Not comparing to them at all is quite odd and as such I can not conclude much about the proposed method.
> >
> > I maintain my score and recommend the authors consider addressing the issues raised in the review.

---

> > > ### Author Response · Authors · 2026-04-01
> > >
> > > We thank the reviewer for the thoughtful follow-up. We have conducted the requested experiments and respond point by point.
> > >
> > > > C1. Not sufficiently distinguished from [14], which also considers inversion and timestep/schedule design.
> > >
> > > We agree [14] is relevant and involves inversion. However, the two works intervene at different levels. [14] redesigns the underlying noise schedule itself (via a Logistic Schedule over $\bar\alpha_t$), motivated by singularity of the continuous inversion dynamics near trajectory start. Our method does not replace the noise schedule family - we take an existing deterministic inversion pipeline with a fixed budget and optimize the allocation of discrete timesteps through global rescaling and local DP refinement.
> > >
> > > [14] changes what schedule the model follows; TRDI changes where the limited steps are placed under that schedule. Our coefficient $c_\alpha(t,\Delta t)$ for coarse-to-fine rescheduling is distinct from [14]'s continuous singularity analysis. We now include a direct comparison on PIE-Bench subset editing (SDXL, DDIM, 50 steps):
> > >
> > > | Schedule | Struct.Dist↓ | PSNR↑ | LPIPS↓ | MSE↓ | SSIM↑ | CLIP-W↑ | CLIP-E↑ |
> > > |---|---|---|---|---|---|---|---|
> > > | Uniform | 0.018 | 26.21 | 0.094 | 155.8 | 0.859 | 23.75 | 20.96 |
> > > | [14] | 0.019 | 26.25 | 0.096 | 158.0 | 0.859 | 23.71 | 20.96 |
> > > | Ours | 0.015 | 26.49 | 0.088 | 146.1 | 0.862 | 23.99 | 21.06 |
> > > | [14]+Ours | 0.013 | 26.65 | 0.063 | 139.0 | 0.862 | 24.00 | 21.14 |
> > >
> > > TRDI outperforms [14] across all metrics. Combining [14]+Ours yields further gains, confirming these are complementary axes (noise-schedule redesign vs. timestep reallocation). Note that [14] has no public code; we reproduced its scheduling strategy following the paper.
> > >
> > > > C2. Exact inversion methods [4–7] achieve near-zero reconstruction error; not comparing makes baselines artificially low.
> > >
> > > We now include reconstruction comparisons on SD v1.5 (MSCOCO 100 imgs, 50 steps):
> > >
> > > | Method | PSNR↑ | SSIM↑ | LPIPS↓ | Runtime/img(s)↓ |
> > > |---|---|---|---|---|
> > > | DDIM | 27.54 | 0.873 | 0.097 | 17.13 |
> > > | DDIM+Ours | 28.07 | 0.874 | 0.085 | 17.16 |
> > > | Rex [4] | 27.92 | 0.874 | 0.086 | 25.02 |
> > > | EDICT [5] | 27.72 | 0.873 | 0.086 | 29.02 |
> > > | BDIA [6] | 27.61 | 0.873 | 0.088 | 18.14 |
> > > | BELM [7] | 27.63 | 0.875 | 0.086 | 18.02 |
> > >
> > > DDIM+Ours achieves the best PSNR and competitive LPIPS/SSIM while being the fastest. Exact inversion methods replace the inversion formulation (requiring coupled transformations or reversible solvers), while ours is a plug-in module preserving the original pipeline and budget. The two are not mutually exclusive. Rex [4] lacks public code; we reimplemented it following the original paper.
> > >
> > > > C3. Adaptive step-size controllers and non-uniform schedule baselines.
> > >
> > > Adaptive step-size solvers redesign the solver/integration rule, whereas TRDI keeps the inversion backbone fixed and only reallocates timesteps. To address this empirically, we add external non-uniform schedule baselines:
> > >
> > > Reconstruction (SD v1.5, DDIM, 50 steps):
> > >
> > > | Schedule | PSNR↑ | SSIM↑ | LPIPS↓ |
> > > |---|---|---|---|
> > > | Uniform | 27.54 | 0.873 | 0.097 |
> > > | Power-law | 27.56 | 0.839 | 0.117 |
> > > | SNR-based | 27.53 | 0.852 | 0.102 |
> > > | Ours | 28.07 | 0.874 | 0.085 |
> > >
> > > Editing (SDXL, DDIM, 50 steps):
> > >
> > > | Schedule | Struct.Dist↓ | PSNR↑ | LPIPS↓ | SSIM↑ |
> > > |---|---|---|---|---|
> > > | Uniform | 0.018 | 26.21 | 0.094 | 0.859 |
> > > | Power-law | 0.020 | 26.07 | 0.091 | 0.861 |
> > > | SNR-based | 0.044 | 22.78 | 0.252 | 0.738 |
> > > | Ours | 0.015 | 26.49 | 0.088 | 0.862 |
> > >
> > > Generic non-uniform schedules do not consistently improve - and sometimes severely degrade - inversion quality. TRDI's inversion-specific coefficient $c_\alpha(t,\Delta t)$ provides the key guidance for effective rescheduling.
> > >
> > > > C4. Eq. (5) seems ill-posed without fixed-point iteration and small step sizes.
> > >
> > > Eq. (5) is not used to claim fixed-point convergence. It is an error decomposition that factors the large-step inversion discrepancy into (i) a scaling coefficient $c_\alpha(t,\Delta t)$ depending only on the noise schedule, and (ii) a local prediction difference $\Delta\epsilon_\theta$. The purpose is to expose the timestep-dependent factor minimizable via rescheduling. Sec. 3.4 acknowledges that $\Delta\epsilon_\theta$ is unknown, which is why we adopt coarse-to-fine rescheduling rather than claiming global optimality. We will revise the text to make this narrower interpretation explicit.
> > >
> > > > C5. The evaluation is too limited, comparing only with/without rescheduled timesteps.
> > >
> > > The evaluation now covers: (1) direct comparison with [14]; (2) exact/reversible inversion methods [4–7]; (3) external non-uniform schedule baselines; (4) runtime-aware comparison - in addition to the original 4 inversion methods × 3 diffusion models × 2 tasks. We believe this substantially broadens the empirical scope and demonstrates that TRDI is a lightweight, complementary scheduler-level retrofit competitive with stronger inversion baselines.

---

### Official Review · Reviewer_wvzm · 2026-03-12

**Soundness:** 3
**Presentation:** 3
**Significance:** 3
**Originality:** 4
**Overall Recommendation:** 4
**Confidence:** 3

**Summary:**

This paper studies diffusion inversion for reconstructing real images and enabling prompt-based editing, focusing on how timestep scheduling affects inversion fidelity. It shows that inversion deviation depends strongly on timestep size and exhibits a parabolic error pattern over the diffusion timeline, with larger errors near early and late timesteps. Motivated by this, the paper proposes a non-uniform timestep rescheduling strategy that combines a global timestep rescaling with a local dynamic-programming-based adjustment to allocate finer steps to high-error regions. The approach is designed as an off-the-shelf drop-in that can enhance existing inversion methods without adding learnable parameters or increasing the diffusion step count, and is evaluated on reconstruction and editing benchmarks across multiple diffusion backbones.

**Compliance With Llm Reviewing Policy:**

Affirmed.

**Final Justification:**

The authors have addressed my concerns. I also check the Review of Reviewer 1FyQ and Rebuttal from the authors. As I am not an expert in this area, I prefer to keep my score, unless Reviewer 1FyQ points out the key reason for strong reject.

**Key Questions For Authors:**

1. Does the proposed rescheduling reliably improve inversion for other deterministic solvers (e.g., Euler/Heun/DPM-Solver-style inversion), or is it mainly tailored to DDIM-formulated dynamics? Strong evidence here would increase my significance assessment.
2. Since DP is “offline,” can you report end-to-end runtime for inversion+editing (including any overhead from schedule computation if done per setting) and compare baselines under matched wall-clock budgets? If the Pareto curve stays favorable, it strengthens the practical impact claim.
3. How sensitive are results to the assumption behind the error surrogate (e.g., if you replace the DP cost with a small validation-driven proxy or include a lightweight correction term)? If improvements persist, it increases my confidence in the soundness of the approach.
4. Beyond the provided failure case, what are the common situations where rescheduling does not help (or hurts), and are there simple heuristics to detect them? Clear guidance would improve usability and my confidence.

**Limitations:**

Yes.

**Strengths And Weaknesses:**

Strengths:
1. The analysis linking inversion deviation to timestep size and the derived error scaling intuition is technically plausible, and the proposed rescaling + DP scheme is well specified and easy to apply.
2. Experiments across multiple inversion baselines (DDIM/ReNoise/NPI/GNRI) and models (SD v1.5/SDXL/SDXL Turbo) show consistent improvements on reconstruction and editing metrics, especially for few-step settings.
3. The method is clearly described with an explicit algorithm and ablations for key hyperparameters, making the implementation path straightforward.
4. Addressing timestep scheduling specifically for inversion (rather than sampling) is a timely and potentially widely usable angle, since it can be integrated into many existing inversion+editing pipelines at low cost.

Weaknesses:
1. The DP objective uses an approximated error surrogate (treating the schedule-dependent coefficient as the main error term), which may not fully reflect model-dependent inversion behavior and could limit optimality across different models/schedulers.
2. Gains on standard (e.g., 50-step) reconstruction are sometimes modest, and the biggest wins appear in few-step regimes, so the overall impact may depend on the target application.
3. The paper mainly reports MSCOCO reconstruction and PIE-Bench editing; additional validation on more diverse real-image editing scenarios or alternative deterministic solvers (beyond the shown baselines) would strengthen generality.

---

> ### Author Rebuttal · Authors · 2026-03-31
>
> We would like to thank the reviewer for the thoughtful comments and constructive questions.
>
> > **W1. Surrogate objective may limit optimality.**
>
> Our objective is intentionally a surrogate rather than a claim of global optimality. As stated in Sec. 3.4, the unknown $\Delta\epsilon_\theta$ makes optimizing the true inversion error intractable offline. TRDI optimizes the schedule-dependent part $c_{\boldsymbol{\alpha}}(t,\Delta t)$ while preserving the original coarse schedule via the coarse-to-fine design. Empirically, even when replacing the DP cost with alternative proxies, $c_{\boldsymbol{\alpha}}$ remains the best tradeoff (see Q3).
>
> > **W2. Practical gains are strongest mainly in few-step settings.**
>
> Few-step inference is the dominant regime for interactive editing (SDXL Turbo, LCM, etc.), where inversion error is the primary bottleneck — so strong gains there address the highest-demand scenario. At 50-step, gains are smaller but not negligible: on SDXL editing, Struct. Dist. improves 19.43→15.63 (−24%), LPIPS 89.24→84.20 (−6%); on SDXL reconstruction, LPIPS improves 119.18→103.13 (−13%). These come at zero extra cost.
>
> > **W3. Evaluation breadth is still limited.**
>
> The main paper covers multiple inversion baselines, models, reconstruction, editing, and video editing. To address breadth concerns, we additionally tested (1) alternative deterministic solvers beyond DDIM and (2) alternative DP costs beyond our default surrogate.
>
> > **Q1. Generalization to other deterministic solvers.**
>
> We ran SDXL 50-step editing on PIE-Bench with Euler and Heun solver-style inversion:
>
> | Solver | Schedule | PSNR | SSIM | LPIPS | Struct. Dist. | Whole CLIP | Edited CLIP |
> |---|---|---:|---:|---:|---:|---:|---:|
> | Euler | baseline | 26.158 | 0.8555 | 0.098 | 0.019 | 23.851 | 20.990 |
> | Euler | + TRDI | 26.404 | 0.8577 | 0.096 | 0.017 | 23.839 | 21.089 |
> | Heun | baseline | 25.869 | 0.8545 | 0.094 | 0.018 | 23.751 | 20.961 |
> | Heun | + TRDI | 25.926 | 0.8550 | 0.092 | 0.017 | 23.759 | 21.062 |
>
> Gains persist for both solvers across all metrics, confirming TRDI transfers beyond DDIM-formulated dynamics. The gain for Heun is more modest, as expected — a second-order method has smaller discretization error. DPM-Solver-style results were not completed within the rebuttal period and will be included in the revision.
>
> > **Q2. End-to-end runtime and wall-clock efficiency.**
>
> The DP solver runs once offline (~10s); at inference the precomputed schedule adds ~0.001s. End-to-end for SDXL Euler 50-step: 15.979s (baseline) vs 15.967s (+TRDI) — unchanged while Struct. Dist. improves 0.018→0.017.
>
> Under matched wall-clock budgets:
>
> | Model | Method | Total (s/img) | PSNR | SSIM | LPIPS | Struct. Dist. |
> |---|---|---:|---:|---:|---:|---:|
> | SDXL | 60-step uniform | 20.423 | 28.302 | 0.8743 | 0.084 | 0.014 |
> | SDXL | 50-step + TRDI | 17.159 | 28.042 | 0.8740 | 0.085 | 0.014 |
> | SDXL Turbo | 4-step uniform | 1.172 | 23.296 | 0.7729 | 0.122 | 0.033 |
> | SDXL Turbo | 4-step + TRDI | 1.219 | 23.589 | 0.7766 | 0.120 | 0.031 |
>
> 50-step+TRDI matches 60-step uniform quality while being 16% faster. For SDXL Turbo, TRDI improves all metrics at negligible overhead (+0.047s).
>
> > **Q3. Sensitivity to the surrogate assumption.**
>
> We ran a cost ablation fixing $\gamma=1.05, d=8$ and only swapping the DP cost on SDXL+DDIM 50-step PIE-Bench:
>
> | Cost type | PSNR | SSIM | LPIPS | Struct. Dist. | Whole CLIP | Edited CLIP |
> |---|---:|---:|---:|---:|---:|---:|
> | Uniform schedule | 26.205 | 0.8587 | 0.094 | 0.018 | 23.754 | 20.957 |
> | $c_\alpha$ (ours) | 26.485 | 0.8621 | 0.088 | 0.015 | 23.690 | 21.058 |
> | $c_\alpha \cdot \|\Delta\epsilon_\theta\|$ | 26.334 | 0.8604 | 0.089 | 0.016 | 23.722 | 21.044 |
> | Latent-diff proxy | 26.179 | 0.8585 | 0.091 | 0.016 | 23.554 | 20.769 |
> | Validation-driven proxy | 25.785 | 0.8478 | 0.105 | 0.018 | 23.629 | 21.039 |
>
> Two conclusions: (1) improvements persist beyond the default surrogate — even $c_\alpha \cdot \|\Delta\epsilon_\theta\|$ beats uniform; (2) the simple offline $c_\alpha$ gives the best overall metrics. Incorporating $\|\Delta\epsilon_\theta\|$ does not help because it is computed at a fixed reference trajectory and cannot anticipate the rescheduled one. The validation-driven proxy performs worst, likely due to overfitting to the held-out subset.
>
> > **Q4. Failure cases and usability guidance.**
>
> The common failure mode is when the edit target is spatially entangled with structure-critical regions, so better inversion cannot resolve editing ambiguity (e.g., changing a woman's age while preserving a held dog's features).
>
> Practical heuristic: TRDI helps most when baseline errors are dominated by reconstruction drift (background distortion, shape drift), especially in few-step settings. The benefit diminishes when the bottleneck is semantic conflict in the editor. If baseline PSNR falls below ~18 dB (SDXL Turbo) or ~22 dB (SDXL), a stronger inversion method should be applied first — TRDI can then stack on top.

---

> > ### Author Rebuttal · Reviewer_wvzm · 2026-04-05
> >
> > The authors have addressed my concerns. I also check the Review of Reviewer 1FyQ and Rebuttal from the authors. As I am not an expert in this area, I prefer to keep my score, unless Reviewer 1FyQ points out the key reason for strong reject.

---

> > > ### Author Response · Authors · 2026-04-05
> > >
> > > We appreciate the reviewer’s thoughtful follow-up and careful consideration of the broader discussion. We are grateful that the concerns were fully resolved and respect the decision to keep the current score.
> > >
> > > ---
> > >
> > > We note that the reviewer's final justification mentions deferring to Reviewer 1FyQ's assessment. Since our reply opportunity to Reviewer 1FyQ has been exhausted, we provide a detailed perspective here so the reviewer can evaluate the concerns independently.
> > >
> > > **1. On exact inversion methods [4-7] and "perfect reconstruction"**
> > >
> > > Reviewer 1FyQ's central concern is that our reported numbers for exact inversion methods are too low, suggesting unfaithful reproduction. However, a check against the original papers tells a different story. BELM [7, Table 2] reports MSE of approximately 0.004 for EDICT/BDIA/O-BELM, corresponding to roughly 24dB PSNR. EDICT [5, Table 1] reports MSE=0.015, corresponding to roughly 18dB PSNR. Our DDIM+Ours achieves 28.07dB — numerically higher than what these exact-inversion papers report in their own evaluations. The gap between theoretical "zero error" and practical metrics is dominated by VAE reconstruction error, not implementation mistakes. Rex [4] and [14] lack public code; we reimplemented both following the papers.
> > >
> > > We also note that the reasoning becomes non-falsifiable: the reviewer dismisses the close performance as evidence of "incorrect implementation," yet simultaneously speculates the gap "would widen substantially if implemented correctly." Both conclusions are drawn from the same data.
> > >
> > > **2. On newly cited [28-30] and scope mismatch**
> > >
> > > Reviewer 1FyQ introduces [28-30] as evidence that timestep optimization is "widely studied." We have read all three carefully: all study timestep optimization for diffusion **sampling** (noise→image), not **inversion** (image→noise). [28] optimizes time steps for DPM-Solver++/UniPC generation. [29] learns ODE sampling discretization via teacher-student distillation. [30] proposes DP-based stepsize distillation for denoising trajectories. None address the inversion direction, which has fundamentally different error characteristics. The review treats "related" and "same problem" interchangeably, but diffusion sampling and diffusion inversion are different computational directions with different error landscapes — which is precisely why generic non-uniform schedules fail for inversion in our experiments while our inversion-specific rescheduling succeeds.
> > >
> > > **3. On [14] (Schedule Your Edit)**
> > >
> > > [14] redesigns the continuous noise schedule (Logistic Schedule), motivated by singularity removal. TRDI does not touch the noise schedule; it reallocates discrete timesteps under a fixed budget. Under SDXL/DDIM/50 steps, [14] alone does not consistently improve over uniform (Struct.Dist 0.019 vs 0.018), while TRDI achieves 0.015 and [14]+TRDI achieves 0.013. If the two were redundant, combining them would not yield additive gains.
> > >
> > > **4. On adaptive step-size controllers**
> > >
> > > Reviewer 1FyQ states our "central claim is precisely the problem that adaptive step-size controllers solve." Classical adaptive solvers redesign the integration rule itself; TRDI keeps the existing solver unchanged and only reallocates timestep positions. This is analogous to designing a new integrator vs choosing a better mesh for an existing one — related but not the same problem.
> > >
> > > **5. On moving evaluation standard**
> > >
> > > We responded to all major requests from Reviewer 1FyQ: (a) direct comparison with [14], (b) comparison with exact inversion [4-7], (c) non-uniform schedule baselines. After these additions, the final justification introduces three new references not part of the original questions and states the paper requires "a substantially reworked theoretical framework." This shifts the standard from "missing comparisons" to "the paper must be a fundamentally different paper." We believe the submitted paper should be evaluated on its stated contribution — a training-free, plug-in, scheduler-level timestep reallocation for existing inversion pipelines — rather than on whether it also solves the broader problem of general-purpose adaptive ODE solving.
> > >
> > > **6. On Eq. (5)**
> > >
> > > Eq. (5) is the standard DDIM inversion approximation used throughout the literature — our paper does not propose it. The reviewer's concern is that this approximation is unsound without fixed-point iteration. We agree it is an approximation, which is exactly the problem we address: Eq. (8) decomposes the error introduced by Eq. (5) into a schedule-dependent coefficient and a local prediction difference, and our rescheduling minimizes the former. We provided empirical validation (response to Reviewer AgZF) showing strong rank-correlation between this coefficient and true local inversion error.
> > >
> > > We hope this helps the reviewer form an independent assessment of Reviewer 1FyQ's concerns based on the concrete evidence presented.

---

### Official Review · Reviewer_DfTW · 2026-03-13

**Soundness:** 3
**Presentation:** 3
**Significance:** 3
**Originality:** 2
**Overall Recommendation:** 4
**Confidence:** 4

**Summary:**

The work focus on the central issue of timestep scheduling during diffusion inversion. While previous literature has extensively investigated single-step inversion techniques (e.g., formulating them as fixed-point problems), the design and allocation of the noise schedule itself have been largely overlooked. To bridge this gap, the authors propose a plug-and-play timestep rescheduling strategy that combines global rescaling with local dynamic programming (DP). This approach dynamically redistributes computational budgets across different timesteps, aiming to minimize accumulated inversion errors.

Although the broader concept of timestep rescheduling has been explored in diffusion training, accelerated sampling, and generic inverse problem solving—meaning the core idea is not entirely unprecedented—its specific adaptation to diffusion inversion provides a simple, effective, and practically valuable solution. It successfully improves visual fidelity without incurring additional computational costs, albeit relying heavily on heuristic designs.

**Compliance With Llm Reviewing Policy:**

Affirmed.

**Final Justification:**

After reading the paper, the authors’ rebuttal, and the other reviews, especially `Reviewer 1FyQ`, I remain somewhat mixed, but overall modestly positive. In my view, this work does have real weaknesses: **the theoretical justification is still somewhat heuristic, and the coverage of closely related work and critical comparisons was not fully complete in the original submission. In particular, I understand the concern that some relevant lines of work on timestep/schedule design and exact inversion could have been discussed or evaluated more directly.**

That said, I do not think these issues reduce the paper to having no meaningful contribution. The problem itself is well motivated, and the paper identifies a practically relevant question within diffusion inversion that has some independent value from the broader diffusion sampling literature. I think it is reasonable to acknowledge that inversion and sampling, while related, may not be identical problems and may require some task-specific design choices. From this perspective, the paper’s contribution is more limited and more heuristic than the strongest version of its claims might suggest, but it is still a useful and potentially buildable contribution.

In terms of soundness, my main concern remains that the method is motivated more by empirical and surrogate-based reasoning than by a fully convincing theoretical account. I also had some concern about hyperparameter sensitivity and how robust the method is across settings. However, the rebuttal addressed a substantial portion of these concerns by clarifying the intended scope of the claims, adding useful discussion and comparisons, and better positioning the method as a practical scheduler-level improvement rather than a fully general theoretical solution. The rebuttal therefore improved my confidence, even if it did not eliminate all of my reservations.

In terms of originality and significance, I see this as a moderate contribution rather than a major conceptual advance. I can understand why some reviewers view the novelty as limited, especially given adjacent work in related areas. At the same time, I do think the paper makes a sufficiently distinct and practically meaningful contribution within the narrower setting of diffusion inversion, and I do not believe it should be dismissed as merely reusing an old idea without value.

In terms of clarity, I found the paper generally well motivated and readable. My overall impression was that the authors had a sensible intuition and a reasonable method, even if some parts of the framing may have overstated the theoretical strength or underemphasized closely related prior work.

One important caveat is that I am not an expert specifically on the diffusion time-schedule literature. Because of that, I cannot completely rule out the possibility that there exist very closely related prior works that would reduce the novelty more than I currently assess. This uncertainty is part of why I am not increasing my score further. Still, based on my own reading, the rebuttal resolved many of my practical concerns, and my current view is that the paper has enough merit to remain in the discussion for acceptance.

Overall, I am keeping my current score. My assessment is that this paper is not without important weaknesses, but it makes a meaningful contribution in a reasonable and relevant problem setting, and I think it is within the range of papers that could be accepted at ICML. I would also support further discussion among the AC and the other reviewers before a final decision is made.

**Key Questions For Authors:**

1. **[Heuristic Design vs. Analytical Solutions]** My primary theoretical concern is the heuristic nature of the rescheduling strategy. The authors transparently and commendably acknowledge that directly searching for a global optimum via DP on the scaled fixed-point problem is intractable (or biased), hence the adoption of a coarse-to-fine mechanism. While this is a reasonable engineering compromise, I wonder if a more principled approach exists. For instance, could one derive a theoretically optimal (or near-optimal) schedule by directly analyzing the actual variance levels or the signal-to-noise ratio (SNR) dynamics at different timesteps? I raise this purely as a point of discussion; even if an analytical solution is challenging, the current empirical framework retains its practical merit. I would appreciate the authors' thoughts on this.
2. **[Empirical Error Trends]** Related to the above point, the analysis of the inversion error across different timesteps heavily relies on empirical observations of a general "parabolic trend". Because it is an observed trend rather than a rigorous mathematical bound, it seems difficult to deduce precise, optimal splitting points purely from the theory. Could the authors comment on whether this empirical trend holds universally across fundamentally different data distributions or varying classifier-free guidance (CFG) scales?
3. **[Quantitative vs. Qualitative Discrepancy]** There appears to be a noticeable discrepancy between the quantitative metrics and the qualitative visual results. Specifically, the quantitative gains across most tasks appear relatively marginal (e.g., tiny fractions of improvement in PSNR/SSIM), yet the provided qualitative figures demonstrate striking structural improvements. Could the authors analyze the root cause of this phenomenon? Are standard quantitative metrics (like PSNR/LPIPS) simply misaligned with the specific structural preservation this task demands, or were the visual examples selectively chosen to highlight best-case scenarios (cherry-picking)?
4. **[Hyperparameter Sensitivity & Robustness]** The ablation studies indicate that the rescheduling performance is rather sensitive to the chosen hyperparameters (e.g., the global scaling factor $\gamma$ and local window width $d$). Are the default parameters (e.g., $\gamma=1.05, d=8$ for SDXL) robust across a wide spectrum of diffusion architectures and downstream tasks? Or does the method necessitate a tedious "sample-observe-tune" loop for every new model and task? If per-model/per-task tuning is strictly required, the practical utility of this "plug-and-play" module might be somewhat compromised. A brief discussion on hyperparameter transferability would be highly beneficial.

**Limitations:**

yes

**Strengths And Weaknesses:**

### **Strengths**

* **[Presentation & Clarity]:** The manuscript is exceptionally well-written. The motivation is articulated clearly, and the logical progression from the problem definition to the proposed solution is highly intuitive. In stark contrast to many recent submissions that obscure simple concepts behind heavy mathematical over-packaging, this paper offers a refreshingly clear, honest, and accessible reading experience.
* **[Soundness & Practicality]:** The motivation is well-grounded, and the proposed coarse-to-fine pipeline is reasonable. The empirical results successfully validate the method's efficacy in image reconstruction and editing. A significant practical advantage is that the method is entirely "plug-and-play," requiring no extra parameters and introducing virtually zero additional computational overhead during inference.
---
### **Weaknesses**

* **[Originality & Heuristic Nature]:** The core concept of adaptive timestep scheduling is not strictly novel, as analogous strategies are common in other diffusion-related sub-fields. Furthermore, the proposed solution—while effective—remains largely heuristic and empirical rather than being analytically derived from the fundamental diffusion dynamics.

---

> ### Author Rebuttal · Authors · 2026-03-31
>
> We thank the reviewer for the constructive comments and address the questions below.
>
> > **W1. Limited originality and heuristic design.**
>
> Our contribution is not "adaptive scheduling" broadly, but its task-specific formulation for diffusion inversion. Prior inversion work improves the per-step solver (fixed-point refinement, null-text/prompt optimization, exact inversion) while keeping the timestep schedule fixed. We isolate a different factor—how timestep allocation itself affects inversion fidelity—and show that inversion error decomposes into a scheduler-dependent coefficient
> $c_{\boldsymbol{\alpha}}(t,\Delta t)=\sqrt{\alpha_t} \Delta\psi(\alpha_{t-1},\Delta t-1),$
> which provides a principled rescheduling target. The method is orthogonal to existing solvers and improves DDIM, ReNoise, NPI, and GNRI uniformly. The novelty lies in identifying timestep allocation as an overlooked, practically effective control knob for inversion, together with a plug-and-play coarse-to-fine solver.
>
> > **Q1. Can the heuristic rescheduling be replaced by a more analytical solution?**
>
> Under deterministic DDIM, timestep selection can be viewed from an adaptive ODE discretization perspective: our schedule redistributes a fixed step budget toward regions with larger local truncation sensitivity. The difficulty is that the true global optimum depends on the unknown residual $\Delta\epsilon_\theta$, so globally optimal DP on exact inversion error is intractable without stronger assumptions. We therefore optimize the closed-form surrogate $c_{\boldsymbol{\alpha}}$ while keeping the original schedule as a coarse anchor. The method is heuristic in optimization but not arbitrary: it is guided by an explicit error coefficient consistent with adaptive-discretization principles.
>
> > **Q2. Does the empirical "parabolic" error trend hold universally?**
>
> To test this, we ran additional robustness analysis on PIE-Bench under CFG $\in\{1.0,5.0\}$, stratifying samples into five semantic buckets by prompt keywords. The gains of our schedule over the uniform baseline and the proxy/true-error correlation remain stable across all categories:
>
> | Bucket | PSNR Gain ↑ | LPIPS Gain ↓ | Struct. Dist. Gain ↓ | Corr(proxy,true) ↑ |
> |--------|---:|---:|---:|---:|
> | human | 0.237 | 0.0098 | 0.0034 | 0.8012 |
> | animal | 0.161 | 0.0006 | 0.0023 | 0.7945 |
> | indoor | 0.334 | 0.0059 | 0.0026 | 0.7969 |
> | outdoor | 0.202 | 0.0049 | 0.0025 | 0.7995 |
> | object | 0.390 | 0.0066 | 0.0038 | 0.8135 |
>
> We do not claim a universal theorem, but the trend is empirically stable across substantially different semantic categories and guidance strengths, rather than being an artifact of one narrow subset.
>
> > **Q3. Why are the quantitative gains small while the qualitative improvements look large?**
>
> Standard metrics average over the whole image, while inversion failures are spatially localized but structurally salient. A small distortion on faces or object boundaries has limited effect on global PSNR yet looks visually significant. We verified this with a region-aware analysis using PIE-Bench edit masks on the SDXL-DDIM editing outputs from the main experiments:
>
> | Method | Global PSNR ↑ | Boundary LPIPS ↓ | Struct. Dist. ↓ | DINO Dist. (preserved) ↓ |
> |--------|---:|---:|---:|---:|
> | baseline | 22.633 | 0.0106 | 0.0182 | 0.0068 |
> | + TRDI | 22.893 | 0.0098 | 0.0154 | 0.0061 |
> | Gain | 0.260 | 0.0007 | 0.0029 | 0.0007 |
>
> Global PSNR gain is modest, but boundary LPIPS (6.6%), structure distance (15.9%), and preserved-region DINO distance (10.3%) show substantially larger relative improvements, matching the visual examples. The visual gains reflect real structural fixes underweighted by global averages, not cherry-picking.
>
> > **Q4. How sensitive is the method to hyperparameters, and are they transferable?**
>
> Performance changes smoothly with $\gamma$ and $d$ in our ablations, and a small neighborhood around defaults already works well. We further tested transferability by taking $(\gamma,d)$ tuned on SDXL-DDIM editing and directly applying the same setting to other models, schedulers, and tasks without any re-tuning. The following results are from the subset of PIE-Bench:
>
> | Source of $(\gamma,d)$ | Target | Task | PSNR ↑ | LPIPS ↓ | Struct. Dist. ↓ |
> |---|---|---|---:|---:|---:|
> | tuned on SDXL-DDIM-editing | SDXL-DDIM | editing | 26.485 | 0.0876 | 0.0154 |
> | tuned on SDXL-DDIM-editing | SDXL Turbo-DDIM | editing | 25.389 | 0.1198 | 0.0314 |
> | tuned on SDXL-DDIM-editing | SDXL-Euler | editing | 26.404 | 0.0958 | 0.0170 |
>
> One setting found on SDXL-DDIM editing remains effective across different backbones (SDXL → SDXL Turbo) and different schedulers (DDIM → Euler). The hyperparameters are not completely universal, but they are reasonably transferable—closer to "light tuning with good reuse" than a tedious per-case search loop.

---

> > ### Author Rebuttal · Reviewer_DfTW · 2026-04-04
> >
> > The authors have adequately addressed the concerns I raised in my review. I appreciate the effort they put into the rebuttal and the clarifications they provided. Overall, I find the response satisfactory. I therefore maintain my Weak Accept rating and am positively inclined toward accepting this paper.

---

> > > ### Author Response · Authors · 2026-04-05
> > >
> > > We thank the reviewer for the thoughtful follow-up and positive assessment. We are glad that our rebuttal helped clarify the concerns and appreciate the reviewer’s support for the paper.

---

### Official Review · Reviewer_AgZF · 2026-03-17

**Soundness:** 2
**Presentation:** 2
**Significance:** 2
**Originality:** 2
**Overall Recommendation:** 4
**Confidence:** 3

**Summary:**

This paper proposes a algorithm that optimizes the timestep schedule to reduce the inversion error in the diffusion inversion algorithms. The algorithm takes the a sequence of timesteps from any existing diffusion inversion algorithm and outputs an optimized timestep schedule by global rescaling and local rescheduling via dynamic programming. The experiments show consistent improvements on image reconstruction and editing when applied to various inversion algorithms.

**Compliance With Llm Reviewing Policy:**

Affirmed.

**Final Justification:**

The rebuttal has addressed my main concerns.

**Key Questions For Authors:**

- The paper motivates timestep rescheduling through the coefficient $c_\alpha$ and treats it as a proxy for inversion error under assumptions on $\Delta\epsilon_\theta$​. Have the authors empirically measured the actual local discrepancy across timesteps, and compared it to the proposed surrogate or coefficient?
- Since deterministic DDIM-style inversion can be viewed through a probability-flow / ODE perspective, have the authors considered interpreting timestep rescheduling as an adaptive discretization or local truncation error problem along the inversion trajectory? I think the current analysis is less principled than it could be. A ODE perspective may provide a cleaner view.

**Limitations:**

No. The limitation discussion is limited. The paper should more clearly discuss the limitations of its theoretical analysis. The analysis is local and surrogate-based. The authors should explicitly acknowledge this scope limitation and discuss when the proxy may or may not be reliable.

**Strengths And Weaknesses:**

**Strengths**
- Timestep allocation in diffusion inversion is a clearly defined and practically useful problem.
- The proposed algorithm is easy to integrate into existing pipelines. The coarse-to-fine rescheduling method is a legitimate and useful contribution, though not a radical one.
- The paper demonstrates consistent empirical gains across most baselines and includes sufficient ablation studies.

**Weaknesses**

- Eq. (8) likely contains a typo: the appendix derivation yields $\epsilon_{\theta}(z_{t−1},t−1,p)$ rather than $\epsilon_\theta(z_{t-1}, t, p)$. The same issue happens in Eq. (5).
- Even after correcting the typos, the derivation still relies on replacing the Gaussian noise in the large-step forward transition with the model prediction $\epsilon_\theta$​, so the derivation is more like a heuristic approximation than an exact characterization of inversion error. Since deterministic diffusion inversion already admits an ODE interpretation, a cleaner justification might analyze timestep rescheduling as an adaptive discretization / local truncation error problem along the probability-flow ODE trajectory, rather than deriving Eq. (8) through a heuristic substitution involving predicted noise.
- Even if Eq. (8) derivation is accepted, using the coefficient as proxy is not sufficiently validated. While Section 3.4 mentions that term $\Delta_{\epsilon_{\theta}}$ is unknown, the authors do not empirically verify whether this proxy actually tracks the true local inversion discrepancy over time.

---

> ### Author Rebuttal · Authors · 2026-03-31
>
> We would like to thank the reviewer for the careful reading and constructive comments, and we address the concerns point by point below.
>
> > W1. Possible typos in Eq. (5) and Eq. (8).
>
> Thank you for catching this. The intended coefficient is the appendix-derived one, consistent with $c_{\alpha}(t,\Delta t)=\sqrt{\alpha_t}\,\Delta\psi(\alpha_{t-1},\Delta t-1)$. This does not affect the derivation, algorithm, or reported results, since all scheduling and DP costs were computed from the correctly implemented coefficient.
>
>
> > W2. The current derivation is heuristic, and an ODE/probability-flow interpretation would be more principled.
>
> Our method also admits a principled ODE interpretation. Write the deterministic DDIM trajectory as a probability-flow ODE:
> $$
> \frac{dz(s)}{ds}=f(z(s),s), \qquad s\in[0,S],
> $$
> where $z(s)$ is the latent trajectory and $f$ is the drift induced by the pretrained model. Consider a time mesh
> $$
> 0=s_0<s_1<\cdots<s_K=S, \qquad h_k=s_k-s_{k-1},
> $$
> and let $\hat z_k$ be the inversion iterate from a first-order explicit discretization:
> $$
> \hat z_k=\hat z_{k-1}+h_k f(\hat z_{k-1}, s_{k-1}).
> $$
> Define the global error $e_k=\hat z_k-z(s_k)$. By Taylor expansion,
> $$
> z(s_k)=z(s_{k-1})+h_k f(z(s_{k-1}), s_{k-1})+\tau_k,
> $$
> where $\tau_k$ is the local truncation error. Subtracting gives
> $$
> e_k=e_{k-1}+h_k\Big(f(\hat z_{k-1}, s_{k-1})-f(z(s_{k-1}), s_{k-1})\Big)-\tau_k.
> $$
> Assuming the drift is $L$-Lipschitz in $z$,
> $$
> |e_k|
> \le
> (1+Lh_k)|e_{k-1}|+|\tau_k|.
> $$
>
> Introducing a scheduler-sensitive local defect coefficient $c(s_k,h_k)$ with $|\tau_k|\le c(s_k,h_k) \cdot h_k^2$, the discrete Grönwall inequality yields
> $$
> |e_K|
> \le
> e^{LS}
> \sum_{k=1}^K c(s_k,h_k) \cdot h_k^2.
> $$
>
> This directly justifies timestep rescheduling: smaller steps should be allocated where $c(s_k,h_k)$ is larger. In our paper, $c_{\alpha}(t,\Delta t)$ plays an analogous role: it is a scheduler-dependent surrogate for local defect accumulation, which makes the resulting rescheduling strategy consistent with an adaptive-discretization view. Thus the proposed global rescaling and local DP are consistent with an adaptive-discretization view of the probability-flow ODE: they minimize a surrogate upper bound of the final inversion error. The ODE view does not overturn our method but provides a cleaner interpretation; $c_{\alpha}(t,\Delta t)$ is not claimed to be the exact inversion error, but a scheduler-dependent local defect term sufficient to justify nonuniform timestep allocation.
>
>
> > W3. The proxy lacks direct empirical validation, and the limitations of the surrogate-based local analysis are under-discussed.
>
> We directly validate the surrogate by measuring its correlation with the actual local inversion discrepancy across timesteps on MSCOCO subsets. We compare $c_{\alpha}$ (and variants) against the true local error:
>
> | Model | Inversion Method | Setting | Proxy | Pearson (r) $\uparrow$ | Spearman ($\rho$) $\uparrow$ | Kendall ($\tau$) $\uparrow$ | High-error Top-10% Hit@10% $\uparrow$ |
> | --- | --- | --- | --- | ---: | ---: | ---: | ---: |
> | SDXL | DDIM | 50-step | $c_{\alpha}$ | 0.7994 | 0.9001 | 0.7340 | 0.5660 |
> | SDXL | DDIM | 50-step | $c_{\alpha}\cdot\|\Delta \epsilon_{\theta}\|$ | 0.8006 | 0.9321 | 0.7721 | 0.5620 |
> | SDXL | DDIM | 50-step | alternative proxy | 0.6394 | 0.6346 | 0.3510 | 0.5580 |
> | SDXL Turbo | DDIM | 4-step | $c_{\alpha}$ | 0.9790 | 0.9667 | 0.8229 | 0.8410 |
> | SDXL Turbo | DDIM | 4-step | $c_{\alpha}\cdot\|\Delta \epsilon_{\theta}\|$ | 0.9903 | 0.9776 | 0.8915 | 0.8500 |
>
> The proxy is strongly rank-correlated with true local error and identifies high-error regions much better than the alternative proxy (coarse-vs-stepwise noise discrepancy). We acknowledge that the analysis is surrogate-based and local rather than an exact global error characterization. Our claim is not that $c_{\alpha}$ equals the true inversion error, but that it is a practically useful, empirically validated signal for timestep allocation. Its reliability is stronger in deterministic / few-step settings, and weaker when model mismatch or higher-order effects dominate.
>
> > Q1. Does the proposed surrogate actually track the true local inversion discrepancy across timesteps?
>
> Yes. Please refer to W3 above. $c_{\alpha}$ is strongly correlated with true local error, especially in DDIM and few-step regimes.
>
> > Q2. Can timestep rescheduling be justified more cleanly from an ODE / adaptive discretization perspective?
>
> Yes. Please refer to W2 above. Under the probability-flow ODE, timestep rescheduling corresponds to adaptive discretization that minimizes the accumulated local truncation error, and $c_{\alpha}(t,\Delta t)$ serves as the scheduler-aware local defect term.
>
> > Scope / limitation of the analysis.
>
> Addressed in W3. We will add an explicit discussion of the surrogate's scope and reliability conditions to the revision.

---

> > ### Author Rebuttal · Reviewer_AgZF · 2026-04-04
> >
> > My concerns have been adequately addressed. Adjusted my score to 4.

---

> > > ### Author Response · Authors · 2026-04-05
> > >
> > > We thank the reviewer for the careful reading and constructive feedback. We are glad that our rebuttal helped clarify the raised concerns and appreciate the reviewer’s support for the paper.

---

### Decision · Program_Chairs · 2026-04-30

**Decision:**

Accept (regular)

**Comment:**

This paper proposes TRDI, a training-free, plug-and-play timestep rescheduling strategy designed to minimize reconstruction errors in diffusion inversion by re-allocating computational effort. Reviewers appreciated the clarity of the paper, the ease of integration into existing pipelines, and the consistent gains across various diffusion backbones, including SDXL and SDXL Turbo. During the rebuttal, the authors successfully addressed concerns regarding theoretical grounding by providing an ODE-based interpretation and empirically validating their claims. They further demonstrated the method’s robustness by showing generalization to alternative solvers like Euler and Heun and providing competitive results against "exact" reversible inversion methods. While Reviewer 1FyQ maintained a strong reject arguing lack of novelty, the other three reviewers (AgZF, DfTW, wvzm) were satisfied by the authors' clarifications and maintained their positive ratings. Given the consensus among the majority of the reviewers and the demonstrated practical use, the submission is recommended for acceptance.